# Convergence Analysis of Wasserstein Proximal Algorithm beyond Geodesic Convexity

## Abstract

The proximal algorithm is a powerful tool to minimize nonlinear and nonsmooth functionals in a general metric space. Motivated by the recent progress in studying the training dynamics of the noisy gradient descent algorithm on two-layer neural networks in the mean-field regime, we provide in this paper a simple and self-contained analysis for the convergence of the general-purpose Wasserstein proximal algorithm without assuming geodesic convexity on the objective functional. Under a natural Wasserstein analog of the Euclidean Polyak-Łojasiewicz inequality, we show that the proximal algorithm achieves an *unbiased* and *dimension-free* linear convergence rate. Our convergence rate improves upon existing rates of the proximal algorithm for solving Wasserstein gradient flows when specialized to strong geodesic convex functionals. We also extend our analysis to the inexact proximal algorithm for geodesically semiconvex objectives. In our numerical experiments, proximal training demonstrates a faster convergence rate than the noisy gradient descent algorithm on two-layer mean-field neural networks.

## 1 Introduction

Minimizing a cost functional over the space of probability distributions has recently drawn widespread statistical and machine learning applications such as variational inference (Lambert et al., 2022; Ghosh et al., 2022; Yao & Yang, 2022), sampling (Wibisono, 2018; Vempala & Wibisono, 2019; Chewi et al., 2024), and generative modeling (Xu et al., 2024; Cheng et al., 2024), among many others. In this work, we consider the following general optimization problem:

$$\min_{\rho \in \mathcal{P}_2(\Theta)} F(\rho), \tag{1}$$

where $F$ is a real-valued target functional defined on the space of probability distributions $\mathcal{P}_2(\Theta)$ with finite second moments on a non-compact domain $\Theta \subset \mathbb{R}^d$. Our motivation for studying this problem stems from analyzing training dynamics of the Gaussian noisy gradient descent algorithm on infinitely wide neural networks, which can be viewed as a forward time-discretization of the mean-field Langevin dynamics (MFLD) (Mei et al., 2019; Hu et al., 2021). Given the connection between sampling and optimization, the continuous-time MFLD is an important example of the Wasserstein gradient flow corresponding to minimizing an entropy-regularized total objective function of large interacting particle systems (cf. Section 2.1).

On the other hand, the Wasserstein gradient flow is conventionally constructed by the following proximal algorithm,

$$\rho_{n+1} \in prox_{F,\xi}(\rho_n) := \arg\min_{\tilde{\rho} \in \mathcal{P}_2(\Theta)} \left\{ F(\tilde{\rho}) + \frac{1}{2\xi} \mathcal{W}_2^2(\tilde{\rho}, \rho_n) \right\}, \tag{2}$$

where $\xi > 0$ is the time-discretization step size. Here, we do not require the proximal operator to have a unique minimizer—our main convergence results in Section 3 for the Wasserstein proximal algorithm hold for any minimizer in $prox_{F,\xi}(\rho_n)$. The Wasserstein proximal algorithm (2) is an iterative *backward time-discretization* procedure for solving (1), and it is also known as the JKO scheme introduced in the seminal work (Jordan et al., 1998). In contrast to various forward-discretization methods such as gradient descent over $\mathcal{P}_2(\Theta)$ and the Langevin sampling algorithms (Durmus et al., 2019; Vempala & Wibisono, 2019;

Chewi et al., 2024), proximal algorithms are often *unbiased* in the sense that their convergence guarantees do not depend on the dimension-dependent discretization error with positive step size and they are often more stable than the forward gradient descent algorithms without strong smoothness condition (Yao & Yang, 2022; Yao et al., 2024; Salim et al., 2020; Cheng et al., 2024). While the proximal algorithms for geodesically convex functionals are well studied in the literature (Naldi & Savaré, 2021; Di Marino et al., 2025), it remains an open question *whether they can maintain similar unbiased and linear convergence guarantees in discrete time beyond the geodesic convexity.* Current work fills this important gap by establishing linear convergence results without assuming geodesic convexity on the objective functional $F$.

Our general quantitative convergence rate for the Wasserstein proximal algorithm offers an alternative training scheme to the noisy gradient descent for two-layer neural networks in the mean-field regime. Specifically, a two-layer neural network is parameterized as

$$f(x; \boldsymbol{\theta}) := \frac{1}{m} \sum_{j=1}^{m} \varphi(\theta_j; x) = \int_{\Theta} \varphi(\theta; x) d\rho^m(\theta), \tag{3}$$

where $\boldsymbol{\theta} = (\theta_1, \theta_2, ..., \theta_m) \in \mathbb{R}^{d \times m}$ and $\rho^m = \frac{1}{m} \sum_{j=1}^{m} \delta_{\theta_j}$ is the empirical distribution of the hidden neuron parameters. The perceptron $\varphi(\theta_j; x)$ in (3) can take the form $\varphi(\theta_j; x) = \sigma(\theta_j^\top x)$ where $\sigma$ is some nonlinear activation function. Given a training dataset $(x_i, y_i)_{i=1}^{N} \sim p(x, y)$ and a convex loss function $l(\cdot)$ (such as the squared loss and logistic loss), the $L^2$-regularized training risk is defined as

$$R(\rho^m) = \frac{1}{N} \sum_{i=1}^{N} l \left( \int_{\Theta} \varphi(\theta; x_i) d\rho^m(\theta), y_i \right) + \lambda \int_{\Theta} \|\theta\|^2 d\rho^m(\theta), \tag{4}$$

where $\lambda > 0$ is the coefficient of $L_2$-regularization. The Gaussian noisy gradient descent algorithm on the $L^2$-regularized training risk can be written as the following stochastic recursion

$$\theta_j^{n+1} = \theta_j^n - \xi \nabla \frac{\delta R}{\delta \rho} (\frac{1}{m} \sum_{\ell=1}^{m} \delta_{\theta_\ell^n})(\theta_j^n) + \sqrt{2\xi\tau} z_{nj}, \tag{5}$$

where $z_{nj}$ are i.i.d. $\mathcal{N}(0, I)$ and $\tau > 0$ represents the Gaussian noise variance. In (5), $\frac{\delta R}{\delta \rho}(\rho)$ is the first variation of $R$ at $\rho$ (cf. Definition A.2). The limiting dynamics of (5) under $m \to \infty$ and $\xi \to 0$ is called the continuous-time MFLD (Hu et al., 2021). Under a uniform log-Sobolev inequality (LSI) assumption (cf. Definition C.1), linear convergence of MFLD to the optimal value of the total objective ($L^2$-training risk plus an entropy term) is established in Nitanda et al. (2022); Chizat (2022), and the noisy gradient descent algorithm is subject to a *dimension-dependent* time-discretization error (Nitanda et al., 2022), which may slow down the convergence.

To remove the time-discretization error, we may instead train the neural network with the Wasserstein proximal algorithm (2). Since such neural network architecture satisfies the uniform LSI which in turn implies a Wasserstein Polyak-Łojasiewicz (PL) inequality (cf. Definition 3.4), our algorithm can achieve an unbiased linear rate of convergence to a global minimum of the total objective.

## 1.1 Contributions

In this work, we give a simple and self-contained convergence rate analysis of the Wasserstein proximal algorithm (2) for minimizing the objective function satisfying a (Wasserstein) PL-type inequality (15) without resorting to any geodesic convexity assumption. Below we summarize our main contributions.

- To the best of our knowledge, current work is among the first works to obtain an *unbiased* and *dimension-free* linear convergence rate of the general-purpose Wasserstein proximal algorithm for optimizing a functional under **merely a PL-type inequality**. Our analysis applied to $\mu$-convex ($\mu > 0$) objective functional along geodesics yields a faster linear convergence rate than the existing literature (Yao & Yang, 2022; Cheng et al., 2024).

- The linear convergence guarantee provides a new training scheme for two-layer wide neural networks in the mean-field regime. Our numerical experiments show a faster training phase (up to particle discretization error) than the (forward) noisy gradient descent method.
- We also analyze the inexact proximal algorithm for geodesically semiconvex objectives under the PL-type inequality.

## 1.2 Literature review

Recently, various time-discretization methods have been proposed for minimizing a functional over a single distribution. Different from the proximal algorithm, some explicit forward schemes that can be seen as gradient descent in Wasserstein space are proposed (Chewi et al., 2020; Liu & Wang, 2016). For example, Chewi et al. (2020) studies a gradient descent algorithm for solving the barycenter problem on the Bures-Wasserstein manifold of Gaussian distributions. The Langevin algorithm, as another forward discretization of Wasserstein gradient flows via its stochastic differential equation (SDE) recursion, is widely used in the sampling literature. Numerous works (Durmus et al., 2019; Vempala & Wibisono, 2019; Chewi et al., 2024) have been devoted to the analysis of the Langevin algorithm under different settings and its variants (Zhang et al., 2023; Wu et al., 2022). However, Langevin algorithms are naturally biased for a positive step size. Salim et al. (2020) introduced a hybridized forward-backward discretization, namely the Wasserstein proximal gradient descent, and proved convergence guarantees for geodesically convex objectives, akin to the proximal gradient descent algorithm in Euclidean spaces. Luu et al. (2024) proposed a semi Forward-Backward scheme to optimize a subclass of non-geodesically convex objectives with a difference-of-convex structure.

**Existing rate analysis for proximal algorithm.** Though convergence rate analysis for Langevin algorithms under strong convexity is well-developed, it is not until recently that the convergence rate of the proximal algorithm on geodesically convex objectives is obtained. One advantage of the proximal algorithm is that it ensures a dimension-independent convergence guarantee directly for any starting distribution. Yao & Yang (2022); Cheng et al. (2024) proved an unbiased linear convergence result for the $\mu$-strongly convex objective. The condition is relaxed to geodesic convexity and quadratic growth of functional in (Yao et al., 2024). However, convergence analysis for non-geodesically convex objective functionals is missing.

**Convergence rate of different time-discretizations under PL-type inequality.** Vempala & Wibisono (2019) obtained a biased linear convergence result for Langevin dynamics under the log-Sobolev inequality (LSI) and smoothness condition. Nitanda et al. (2022) extended this result to MFLD with similar techniques. The proximal Langevin algorithm proposed by Wibisono (2019), attains a biased linear convergence rate under the LSI, while an extra smoothness condition of the second derivative of the sampling function is required. Proximal sampling algorithm (Chen et al., 2022b), assuming access to samples from an oracle distribution, achieves an unbiased linear convergence for sampling from Langevin dynamics under the LSI, while the analysis requires geodesic semiconvexity (cf. Definition A.3). Fan et al. (2023); Liang & Chen (2024) improved the results, however, their focus is still on sampling on a fixed function and cannot be applied to MFLD.

To highlight the distinction between our contributions and existing results from the literature, we make the following comparison in Table 1 between explicit convergence guarantees of the Wasserstein proximal algorithm and Langevin algorithms for optimizing the KL divergence (7). Similar comparison on the convergence rates can be made between our result and the forward time-discretization of MFLD under further assumptions (Nitanda et al., 2022; Chizat, 2022). In Table 1, we would like to make a comment on the proximal convergence rates (i.e., Rows 2 and 3). Note that $\mu$-strong convexity of a general functional $F$ does not directly imply $\mu$-PL inequality, and the rate we obtained for the proximal algorithm in Row 3 of Table 1 does not require the verification of Assumption 2 (cf. Theorem 3.10). Thus, despite the same convergence rate of our results, the analysis of Row 3 in our paper follows Theorem 3.10 and does not directly follow from the analysis of Row 2 (Corollary 3.8).

The remainder of this paper is organized as follows. In Section 2, we provide some background knowledge for the connection between Wasserstein gradient flows and associated Langevin dynamics. In Section 3, we present our main convergence results. In Section 4, we discuss how to apply the proximal algorithm for MFLD

Table 1: Comparison between Langevin (forward discretization) and Wasserstein proximal algorithms for the KL divergence functional with a target distribution $\nu = e^{-f}$.

| Algorithm | Assumptions | Step size | Convergence guarantee at $n$-th iteration |
|---|---|---|---|
| Langevin | $\nu$ is $\mu$-LSI
$f$ is $L$-smooth (on $\Theta$) | $0 < \xi < \dfrac{\mu}{4L^2}$ | $e^{-n\mu\xi}D_{\mathrm{KL}}(\rho_0\|\nu) + \dfrac{8\xi dL^2}{\mu}$
Vempala & Wibisono (2019) |
| Proximal | $\nu$ is $\mu$-LSI
$f$ is semiconvex (on $\Theta$) | $\xi > 0$ | $\dfrac{1}{(1+\mu\xi)^{2n}}D_{\mathrm{KL}}(\rho_0\|\nu)$
Ours, Corollary 3.8 |
| Proximal | $f$ is $\mu$-strongly convex
(on $\Theta$) | $\xi > 0$ | $\dfrac{1}{(1+\mu\xi)^n}D_{\mathrm{KL}}(\rho_0\|\nu) \to \dfrac{1}{(1+\mu\xi)^{2n}}D_{\mathrm{KL}}(\rho_0\|\nu)$
Yao & Yang (2022)   Ours, Theorem 3.10 |

of a two-layer neural network and provide numerical experiments exploring the behavior of the proximal algorithm.

**Notations.** We assume $\Theta = \mathbb{R}^d$ (by default) throughout the paper unless explicitly indicating that it is a compact subset of $\mathbb{R}^d$. Let $\mathcal{P}_2(\Theta)$ be the collection of all probability measures with finite second moment, and $\mathcal{P}_2^a(\Theta) \subset \mathcal{P}_2(\Theta)$ be the subset of absolutely continuous measures. For a measurable map $T : \Theta \to \Theta$, let $T_\# : \mathcal{P}_2(\Theta) \to \mathcal{P}_2(\Theta)$ be the corresponding pushforward operator. For probability measures $\mu$ and $\nu$, we shall use $T_\mu^\nu$ to denote the optimal transport (OT) map from $\mu$ to $\nu$ and $\mathbf{id}$ to denote the identity map. We use $\mathcal{W}_2(\cdot, \cdot)$ to denote the 2-Wasserstein distance. We denote $\partial F(\rho)$ to be the Fréchet subdifferential at $\rho \in \mathcal{P}_2^a(\Theta)$ if exists, $\mathcal{D}(F) := \{\rho \in \mathcal{P}_2(\Theta) \mid F(\rho) < \infty\}$ to be the domain of $F$ that has finite functional value, and $D(|\partial F|)$ to be the domain of $F$ that has finite metric slope, see Lemma 10.1.5 of Ambrosio et al. (2008). We refer to Appendix A for more notions and definitions.

## 2 Preliminaries

In this section, we review the connection between Wasserstein gradient flows and the associated Langevin dynamics.

### 2.1 Wasserstein gradient flows and continuous-time Langevin dynamics

Gradient flows in the Wasserstein space of probability distributions provide a powerful means to understand and develop practical algorithms for solving diffusion-type equations (Ambrosio et al., 2008). For a smooth Wasserstein gradient flow, noisy gradient descent algorithms over relative entropy functionals are often used for space-time discretization via the stochastic differential equation (SDE). Below we illustrate two main Wasserstein gradient flow examples involving the linear and nonlinear Fokker-Planck equations, which model the diffusion behavior of probability distributions.

**Langevin dynamics via the Fokker-Planck equation.** The Langevin dynamics for the target distribution $\nu = e^{-f}$ is defined as an SDE,

$$d\theta_t = -\nabla f(\theta_t)dt + \sqrt{2}dW_t \tag{6}$$

where $W_t$ is the standard Brownian motion in $\Theta$ with zero initialization. It is well-known that, see e.g., Chapter 8 of (Santambrogio, 2015), if the process $(\theta_t)$ evolves according to the Langevin dynamics in (6), then their marginal probability density distributions $\rho_t(\theta)$ satisfy the Fokker-Planck equation

$$\partial_t\rho_t - \Delta\rho_t - \nabla \cdot (\rho_t\nabla f) = 0,$$

which is the Wasserstein gradient flow for minimizing the KL divergence (i.e., the relative entropy)

$$D_{\mathrm{KL}}(\rho\|\nu) = \int_\Theta f(\theta)\rho(\theta)d\theta + \int_\Theta \rho(\theta)\log\rho(\theta)d\theta. \tag{7}$$

If $\nu$ satisfies a log-Sobolev inequality (LSI) with constant $\mu > 0$, i.e., if for all $\rho \in \mathcal{P}_2^a(\Theta)$, we have

$$D_{\mathrm{KL}}(\rho\|\nu) \leq \frac{1}{2\mu} J(\rho\|\nu), \tag{8}$$

where $J(\rho\|\nu) = \int_\Theta \left\|\nabla \log \frac{\rho}{\nu}\right\|^2 d\rho$ is the relative Fisher information, then the continuous-time Langevin $\rho_t$ converges to $\nu$ exponentially fast (Bakry et al., 2013).

**Mean-field Langevin dynamics (MFLD) via the McKean-Vlasov equation.** In an interacting $m$-particle system, the potential energy contains a nonlinear interaction term in addition to $\int_\Theta f d\rho$ in (7). More generally, in the mean-field limit as $m \to \infty$, the nonlinear Langevin dynamics can be described as

$$d\theta_t = -\nabla \frac{\delta R}{\delta \rho}(\rho_t)(\theta_t)dt + \sqrt{2\tau}dW_t, \tag{9}$$

where $R : \mathcal{P}_2(\Theta) \to \mathbb{R}$ is a cost functional such as the $L^2$-regularized training risk of mean-field neural networks in (4) and $\tau > 0$ is a temperature parameter. For a convex loss $l$, the risk $R$ in (4) has linear convexity. Process evolving according to (9) solves the following McKean-Vlasov equation (Yao et al., 2022),

$$\partial_t \rho_t - \tau \Delta \rho_t - \nabla \cdot (\rho_t \nabla \frac{\delta R}{\delta \rho}(\rho_t)) = 0, \tag{10}$$

which is the Wasserstein gradient flow of the entropy-regularized total objective,

$$F_\tau(\rho) = R(\rho) + \tau \int_\Theta \rho \log \rho. \tag{11}$$

Similarly, as in the linear Langevin case if the proximal Gibbs distribution of $\rho$ satisfies a uniform LSI (cf. Definition C.1), then MFLD converges to the optimal value exponentially fast in continuous time (Chizat, 2022; Nitanda et al., 2022) and, in the case of infinite-width neural networks in mean-field regime, it is subject to a dimension-dependent time-discretization error (Nitanda et al., 2022).

## 3 Convergence rate analysis

In this section, we first introduce a natural PL inequality in the Wasserstein space and then provide the convergence rate analysis for the Wasserstein proximal algorithm under such a weak assumption. Then, we shall extend our analysis to the inexact proximal algorithm setting. Throughout this section, we make the following regularity assumption.

> **Assumption 1** (Regularity assumption). The functional $F : \mathcal{P}_2(\Theta) \to (-\infty, +\infty]$ satisfies
> (1) Proximal algorithm (2) admits a minimizer for any $\rho \in \mathcal{P}_2^a(\Theta)$ and $\xi > 0$;
> (2) $\mathcal{D}(F) \subset \mathcal{P}_2^a(\Theta)$.

Assumption 1 ensures that the proximal algorithm admits a minimizer in $\mathcal{P}_2^a(\Theta)$. Weak lower semicontinuity of $F$ is a sufficient condition for the existence of a minimizer (cf. Lemma B.2), and some examples of weakly lower semicontinuous functions are provided in Remark B.3. We refer the reader to Remark B.4 for other conditions that guarantee (1) in Assumption 1.

**Definition 3.1** (Hopf-Lax formula). Let $\xi > 0$ and $\rho_\xi \in prox_{F,\xi}(\rho)$. The Hopf-Lax formula $u(\rho, \xi)$ of a functional $F : \mathcal{P}_2(\Theta) \to \mathbb{R}$ is defined as

$$u(\rho, \xi) := F(\rho_\xi) + \frac{1}{2\xi} \mathcal{W}_2^2(\rho_\xi, \rho). \tag{12}$$

The Hopf-Lax formulation in (12) is also known as the Moreau-Yoshida approximation (Ambrosio et al., 2008). Below, we present a key connection between the time-derivative of the Hopf-Lax semigroup and the squared Wasserstein distance between the proximal and the initial point.

**Lemma 3.2.** *Let $\rho \in \mathcal{D}(F)$. Under Assumption 1, we have*

$$\partial_\xi u(\rho, \xi) = -\frac{1}{2\xi^2} \mathcal{W}_2^2(\rho_\xi, \rho) \tag{13}$$

*holds for $\xi \in (0, +\infty)$ with at most countable exceptions, where $\rho_\xi$ can be any proximal operator minimizer in (12).*

*Remark* 3.3 (Computation of the proximal operator). To compute $\rho_{n+1}$ in (2), we can reformulate the proximal algorithm into an optimization problem in functional space. Finding $\rho_{n+1}$ is equivalent to finding an optimal transport (OT) map $T$ such that $T_\# \rho_n$ minimizes (2),

$$T_{\rho_n}^{\rho_{n+1}} = \arg \min_{T:\Theta \to \Theta} \left\{ F(T_\# \rho_n) + \frac{1}{2\xi} \int_\Theta \|T(\theta) - \theta\|^2 \, d\rho_n \right\}. \tag{14}$$

### 3.1 A Wasserstein PL inequality

In this subsection, we introduce the PL inequality in Wasserstein space as in (Boufadène & Vialard, 2023).

**Definition 3.4** (Wasserstein Polyak-Łojasiewicz inequality). For any $\rho \in \mathcal{D}(F)$, the objective functional $F$ satisfies the following inequality with $\mu > 0$,

$$\int_\Theta \left\| \nabla \frac{\delta F}{\delta \rho}(\rho) \right\|^2 \, d\rho \geq 2\mu(F(\rho) - F^*), \tag{15}$$

where $F^* = F(\rho^*)$ and $\rho^*$ is any global minimizer of $F$. We call (15) a Wasserstein PL inequality.

The Wasserstein PL inequality generalizes the classical PL inequality in Euclidean space $\|\nabla f(\theta)\|^2 \geq 2\mu(f(\theta) - f(\theta^*))$ where $f : \Theta \to \mathbb{R}$ (Karimi et al., 2016), with a key technical difference that functional on Wasserstein space may not have very good differentiability structure (cf. the discussion after Assumption 2 below). For KL divergence, the Wasserstein analog of the Euclidean PL inequality is LSI in (8), which is almost exclusively used to study the linear convergence for KL divergence-type objective functionals. With a convex function $f$, the quadratic growth of $f$ implies the PL inequality in Euclidean space (Karimi et al., 2016) and the quadratic growth of its KL objective implies LSI (Yao et al., 2024). Previous works show that under certain regularity conditions, the continuous-time dynamics exhibit linear convergence under the Wasserstein PL inequality (Boufadène & Vialard, 2023; Kondratyev et al., 2016; Chizat, 2022). Our paper considers the problem of minimizing a *general functional* in (1), where the convergence analysis of the proximal algorithm is directly based on the Wasserstein PL inequality (15).

**Assumption 2** (Minimum selection on proximal trajectory). For every $\rho \in \mathcal{D}(F)$ and every $\rho_\xi \in prox_{F,\xi}(\rho)$, $\nabla \frac{\delta F}{\delta \rho}(\rho_\xi)$ is a strong subdifferential at $\rho_\xi$ such that

$$\left\| \nabla \frac{\delta F}{\delta \rho}(\rho_\xi) \right\|_{L_2(\rho_\xi)} \leq \left\| \frac{T_{\rho_\xi}^\rho - \mathbf{id}}{\xi} \right\|_{L_2(\rho_\xi)}$$

Some remarks of Assumption 2 are now in order.

First, it follows from Lemma 10.1.2 in (Ambrosio et al., 2008) that $\rho_\xi \in D(|\partial F|)$ and $(T_{\rho_\xi}^\rho - \mathbf{id})/\xi$ is a strong subdifferential of $F$ at $\rho_\xi$ (cf. Definition A.1 in Supplementary Materials). If $\Theta$ is a compact set in $\mathbb{R}^d$, we have $\nabla \frac{\delta F}{\delta \rho}(\rho_\xi) = (T_{\rho_\xi}^\rho - \mathbf{id})/\xi$ due to the existence of the first variation of $\mathcal{W}_2(\cdot, \rho)$ distance for any fixed $\rho \in \mathcal{P}_2(\Theta)$ (cf. Lemma B.5), and thus Assumption 2 automatically holds. Moreover, for non-compact $\Theta$, both MFLD under the conditions of Corollary 3.7 and Langevin dynamics under the conditions of Corollary 3.8 satisfy Assumption 2 since $\nabla \frac{\delta F}{\delta \rho}(\rho_\xi)$ is guaranteed to be the strong subdifferential at $\rho_\xi$ with minimal $L_2(\rho_\xi)$-norm (see proof of Corollary 3.7 and Corollary 3.8).

Next, we shall highlight throughout this paper that *we do not assume the Wasserstein differentiability, which is an overly restrictive assumption*. For instance, the (geodesically convex) negative entropy functional

$\mathrm{Ent}(\rho) = \int \rho \log \rho$ is not Wasserstein differentiable (see Appendix A for definitions) for any $\rho \in \mathcal{P}_2^a(\mathbb{R}^d)$ such that $\mathrm{Ent}(\rho) < \infty$ (cf. Remark 2.26 in Lanzetti et al. (2025)). More precisely, even strong subdifferential may not exist everywhere and $\nabla \frac{\delta F}{\delta \rho}(\rho)$ is not guaranteed to be a strong subdifferential (e.g., it is not $L_2$ integrable even for $\rho \in \mathcal{P}_2^a(\mathbb{R}^d)$ when $F = \mathrm{Ent}$). Thus, we need to rely on the fact that $F$ is subdifferentiable at $\rho_\xi$ and $(T_{\rho_\xi}^\rho - \mathbf{id})/\xi$ is a strong subdifferential of $F$ at proximal $\rho_\xi$. However, we cannot simply relate the strong subdifferential $(T_{\rho_\xi}^\rho - \mathbf{id})/\xi$ to $\nabla \frac{\delta F}{\delta \rho}(\rho_\xi)$ without Assumption 2, which is a key step to prove our main convergence result in Theorem 3.5. Rather, we connect the proximal iterates $\rho_\xi$ (in terms of the strong subdifferential $(T_{\rho_\xi}^\rho - \mathbf{id})/\xi$) and the PL inequality in (15) via the minimal selection principle in Assumption 2.

Moreover, unlike the Euclidean space, strong $\mathcal{W}_2$-geodesic convexity generally does not directly imply the Wasserstein PL inequality (15). The proof of Theorem 3.10 doesn't require $\nabla \frac{\delta F}{\delta \rho}(\rho_\xi)$ to be strong subdifferential and doesn't rely on the Wasserstein PL inequality. It utilizes the strong geodesic convexity and the strong subdifferential $(T_{\rho_\xi}^\rho - \mathbf{id})/\xi$ at $\rho_\xi$ to circumvent relying on Assumption 2.

### 3.2 Convergence rates of exact proximal algorithm

Now, we are ready to state the main theorem of this paper to establish the convergence rate for the Wasserstein proximal algorithm (2).

**Theorem 3.5** (Convergence rate of the exact proximal algorithm under PL inequality). *Under Assumptions 1 and 2, if the objective functional $F$ in (1) satisfies the PL inequality (15), then for any $\xi > 0$, the Wasserstein proximal algorithm (2) satisfies*

$$F(\rho_n) - F^* \leq \frac{1}{(1 + \xi\mu)^{2n}}(F(\rho_0) - F^*). \tag{16}$$

*Remark* 3.6. It is worth mentioning that our Theorem 3.5 extends Theorem 9 in Chen et al. (2022b) from the Euclidean space to the Wasserstein space with similar computations. However, we want to point out that the proof of their Theorem 9 is not rigorous. Specifically, Chen et al. (2022b) verified that the Moreau envelope satisfies the Hamilton-Jacobi (HJ) equation through the chain rule on $f_{t,x}(x_t)$ (see definition therein), where the time-derivative of the trajectory $x_t$ is not well-defined because its uniqueness is not guaranteed. In contrast, our Lemma 3.2 specialized to the Euclidean context can directly lead to satisfaction of the HJ equation, which provides a more rigorous proof than Chen et al. (2022b).

Next, we specialize our general-purpose convergence guarantee for the Wasserstein proximal algorithm to MFLD induced from training a two-layer neural network (3) in the mean-field regime.

**Corollary 3.7** (Wasserstein proximal algorithm on MFLD). *Let $F_\tau$ be the total objective functional in (11). Suppose that the loss function $l(\cdot)$ is either squared loss or logistic loss. If the perceptron $\varphi(\theta; x)$ is bounded by $K$ and $\sup_{\theta,x} \|\nabla_\theta \varphi(\theta; x)\|$ is finite, then for any $\xi > 0$, the Wasserstein proximal algorithm in (2) with the objective functional $F_\tau$ satisfies*

$$F_\tau(\rho_n) - F_\tau^* \leq \frac{1}{(1 + \xi\tau\mu_\tau)^{2n}}(F_\tau(\rho_0) - F_\tau^*),$$
$$\mathcal{W}_2(\rho^*, \rho_n) \leq \sqrt{\frac{2}{\mu_\tau\tau}(F_\tau(\rho_0) - F_\tau^*)} \left(\frac{1}{1 + \xi\mu_\tau\tau}\right)^n. \tag{17}$$

Our Theorem 3.5 can also be applied to derive the convergence rate for backward time-discretized KL divergence (i.e., linear Langevin dynamics).

**Corollary 3.8** (Wasserstein proximal algorithm on Langevin dynamics). *Suppose $\nu = e^{-f}$ satisfies the $\mu$-LSI condition (8) for a potential function $f : \Theta \to \mathbb{R}$. If $f$ is semiconvex, lower semicontinuous, then for any $\xi > 0$, the Wasserstein proximal algorithm in (2) with the KL divergence objective functional $D_{\mathrm{KL}}$ satisfies*

$$D_{\mathrm{KL}}(\rho_n\|\nu) \leq \frac{1}{(1 + \mu\xi)^{2n}}D_{\mathrm{KL}}(\rho_0\|\nu),$$
$$\mathcal{W}_2(\rho_n, \nu) \leq \sqrt{\frac{2}{\mu}D_{\mathrm{KL}}(\rho_0\|\nu)} \frac{1}{(1 + \mu\xi)^n}.$$

We remark that the objective functions in Corollaries 3.7 and 3.8 are linearly convex. In particular, the linear convexity of the MFLD objective plays a crucial role in establishing the entropy sandwich argument (cf. proof of Corollary 3.7), which in turn yields the Wasserstein PL inequality in Corollary 3.7. This raises a natural question: are there examples satisfying Wasserstein PL inequality that are neither linearly convex nor geodesically convex? The following example gives an affirmative answer.

**Corollary 3.9** (A Wasserstein PL example without linear or geodesic convexity). *Let $\Theta = \mathbb{R}$ and fix $c \in (0,1)$. Define the functional $F : \mathcal{P}_2(\mathbb{R}) \to (-\infty, +\infty]$ by*

$$F(\rho) := \begin{cases} (m(\rho) + c\sin(m(\rho)))^2, & \rho \in \mathcal{P}_2^a(\mathbb{R}), \\ +\infty, & \rho \in \mathcal{P}_2(\mathbb{R}) \setminus \mathcal{P}_2^a(\mathbb{R}), \end{cases}$$

*where*

$$m(\rho) := \int_{\mathbb{R}} \theta \, d\rho(\theta).$$

*Then $F$ satisfies Assumptions 1, 2,and $F$ satisfies the Wasserstein PL inequality with constant $\mu = 2(1-c)^2$. However, for $c = 1/2$, the functional $F$ is neither linearly convex nor geodesically convex.*

Next, applying similar proof techniques to the $\mu$-strongly convex objective functional $F$ in (1), we obtain a faster convergence rate than those in existing literature, see Remark 3.11 below. In particular, we can avoid evoking Assumption 2 with careful modifications to our proof.

**Theorem 3.10** (Sharper convergence rates of the exact proximal algorithm: strongly geodesically convex objective). *Under Assumption 1, if the objective functional $F$ in (1) is $\mu$-strongly convex along geodesics, then for any $\xi > 0$, the Wasserstein proximal algorithm (2) satisfies*

$$F(\rho_n) - F^* \leq \frac{1}{(1+\mu\xi)^{2n}}(F(\rho_0) - F^*),$$

$$\mathcal{W}_2(\rho^*, \rho_n) \leq \sqrt{\frac{2}{\mu}(F(\rho_0) - F^*)} \frac{1}{(1+\mu\xi)^n}.$$

*Remark* 3.11. The bound obtained in Theorem 3.10 is sharper than those in (Yao & Yang, 2022) and (Cheng et al., 2024) for $\mu$-strongly convex objective, where the convergence rate for functional value $(1+\mu\xi)^{-n}(F(\rho_0) - F^*)$ is proved. Theorem 3.10 applies to functionals corresponding to interacting particle systems $F(\rho) = \int_\Theta f(\theta)d\rho(\theta) + \int_\Theta w(\theta_1, \theta_2)d\rho(\theta_1)d\rho(\theta_2) + \int_\Theta \log\rho(\theta)d\rho(\theta)$, where both the external force $f(\cdot)$ and interaction potential $w(\cdot, \cdot)$ are both convex and at least one of them is strongly convex (Yao et al., 2024).

### 3.3 Convergence rates of inexact proximal algorithm

Since the trajectory OT map $T_{\rho_n}^{\rho_{n+1}}$ in (14) is learned through data, this subsection investigates the inexact proximal algorithm where numerical errors are allowed in each iteration. Let $\bar{\rho}_n$ be the inexact solution of the proximal algorithm at iteration $n$. We need an additional smoothness assumption on the proximal flow to provide a quantitative analysis for the proximal algorithm when the OT map at each iteration is allowed to be estimated with errors.

> **Assumption 3** (Smoothness of inexact proximal flow). If $\bar{\rho}_n \in \mathcal{C}^1(\Theta)$, then $\bar{\rho}_{n+1} \in \mathcal{C}^1(\Theta)$, for all $n \in \mathbb{N}$.

When the proximal algorithm (2) is initialized with $\rho_0 \in \mathcal{C}^1(\Theta)$, Assumption 3 ensures that the inexact proximal flow $\bar{\rho}_n$ remains $\mathcal{C}^1(\Theta)$. In practice, to optimize over (14) via $T_{\bar{\rho}_n}^{\bar{\rho}_{n+1}}$, the learned OT map is typically restricted to a specific class, such as a normalizing flow (Xu et al., 2024) or a neural network (Yao & Yang, 2022) with a certain structure. Even though this assumption is mainly for technical purposes, we provide Lemma B.8 in the Appendix, showing that restricting the learned OT map to some classes can ensure Assumption 3. Next we define the error $\beta_{n+1} : \Theta \to \Theta$ in estimating the Wasserstein subdifferential in the $(n+1)$-th iterate as,

$$\beta_{n+1} = \xi^{-1}(T_{\bar{\rho}_{n+1}}^{\bar{\rho}_n} - \mathbf{id}) - \nabla\frac{\delta F}{\delta\rho}(\bar{\rho}_{n+1}). \tag{18}$$

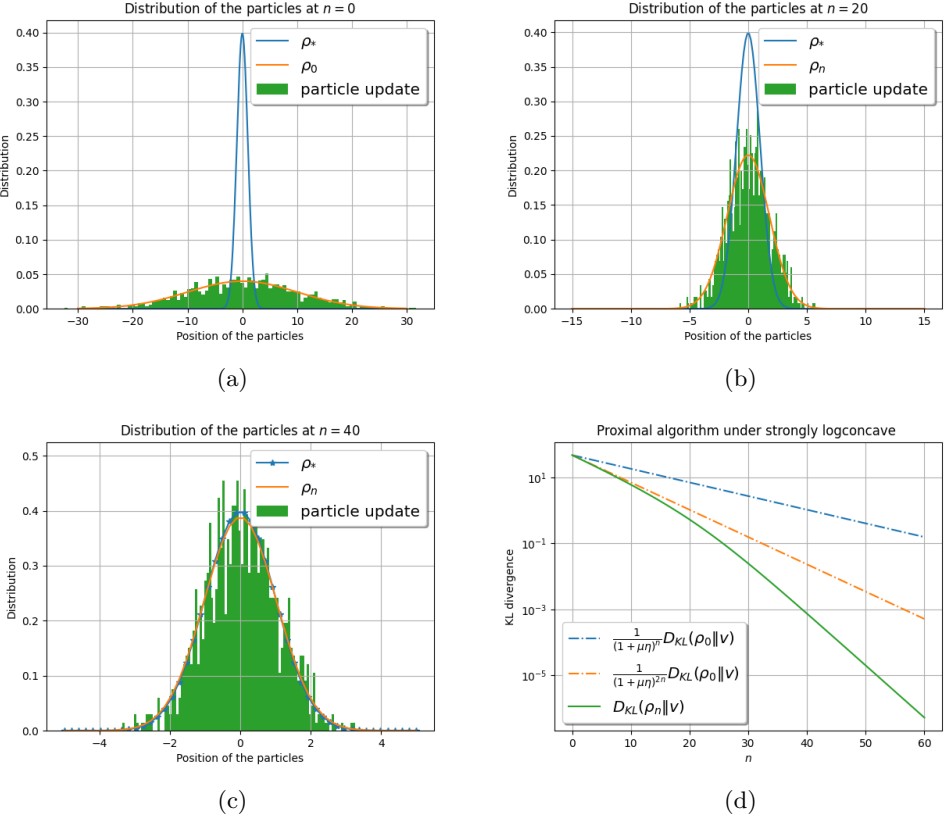

Figure 1: Wasserstein proximal algorithm on the KL divergence with $\nu(\theta) = \exp(-\theta^2/2)$.

The definition (18) of inexact subgradient follows from similar ideas as in (Cheng et al., 2024; Yao & Yang, 2022). Under Assumption 3, if $\overline{\rho}_{n+1}$ is the exact solution of the Wasserstein proximal algorithm (2), then $\beta_{n+1} = 0$ (cf. Lemma B.10). Therefore, we utilize

$$\int_\Theta \|\beta_{n+1}\|^2 d\overline{\rho}_{n+1} \le \epsilon_{n+1}, \tag{19}$$

which can be viewed as measuring the norm of strong subdifferential as in the Euclidean case, to depict the error induced by solving (2) inexactly. Furthermore, we need the additional geodesic semiconvexity assumption apart from PL inequality to control the inexact error, which differs from Section 3.2 that only relies on PL inequality. Examples that satisfy both semiconvexity and PL inequality include the objective of MFLD under the assumptions of Corollary 3.7 and KL divergence objective under the assumptions of Corollary 3.8. Now we are in the position to quantify the impact of numerical errors on the proximal algorithm, which demonstrates how the inexact error impacts the convergence behavior under PL inequality.

**Theorem 3.12** (Inexact proximal algorithm under PL inequality). *Suppose $\overline{\rho}_n \in D(|\partial F|)$ for every $n \in \mathbb{N}$. Under Assumptions 1 and 3, if $F$ is $(-L)$- geodesically semiconvex, $F$ satisfies the PL inequality (15), $\epsilon_n \le \epsilon$ for $n \in \mathbb{N}$ and $0 < \xi \le \frac{1}{L}$, then we have,*

$$F(\overline{\rho}_n) - F^* \le \frac{1}{(1+\mu\xi)^n}(F(\rho_0) - F^*) + \frac{\epsilon(1+\mu\xi)}{2\mu}. \tag{20}$$

We remark that the numeric error bound $\epsilon$ does not have to be small and does not need to converge to zero. We shall explore the behavior of the numeric error through simulations in Section 4.2.

To make our complexity analysis complete, we give algorithmic convergence rates when we have increasing accuracy to solve the proximal subproblems as the algorithm progresses (i.e., $\epsilon_n \to 0$ as $n \to \infty$).

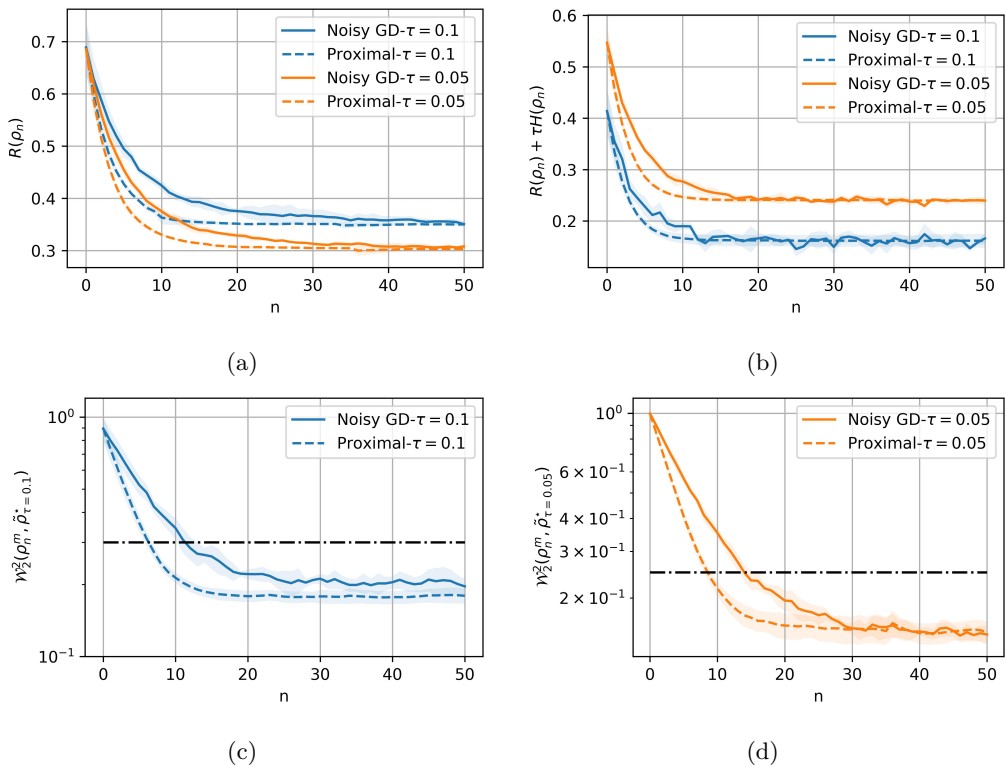

Figure 2: Wasserstein proximal algorithm on MFLD objective. Note that for Figure 2c and Figure 2d, the y-axis is on log scale.

**Corollary 3.13** (Inexact proximal algorithm under PL inequality with increasing accuracy). *Under the assumptions of Theorem 3.12, we have the following. (a) If $\epsilon_n \leq C_{exp}\gamma^n$ with $\gamma \in (0,1)$, then there exists $C_1 = C_1(\mu, \xi, \gamma, C_{exp})$ such that*

$$F(\overline{\rho}_n) - F^* \leq \frac{1}{(1+\mu\xi)^n}(F(\rho_0) - F^*) + C_1 \max\left\{\frac{1}{1+\mu\xi}, \gamma\right\}^{n+1}. \tag{21}$$

*(b) If $\epsilon_n \leq C_{poly}n^{-\zeta}$ with $\zeta > 0$, then there exists $C_2 = C_2(\mu, \xi, \zeta, C_{poly})$ such that*

$$F(\overline{\rho}_n) - F^* \leq \frac{1}{(1+\mu\xi)^n}(F(\rho_0) - F^*) + \frac{C_2}{n^\zeta}. \tag{22}$$

Corollary 3.13 gives a quantitative decay of how the inexact error impacts the convergence behavior under the PL inequality.

## 4 Numerical experiments

In this section, we present two applications of the exact proximal algorithm. The first is on the KL divergence functional, where the particle and the distribution updates can be computed explicitly. The second is on the proximal algorithm training for the regularized objective of two-layer mean-field neural networks.

### 4.1 Linear Langevin dynamics

We apply the proximal algorithm on KL divergence with the target distribution $\nu(\theta) = \exp(-\theta^2/2)$ where $\theta \in \mathbb{R}$. Note that $D_{KL}(\cdot\|\nu)$ is 1-strongly convex along geodesics. When both initialization and the target distributions are Gaussian, problem (14) can be explicitly solved and $\rho_n$ remains Gaussian for every $n$. Closed forms of the particle and distribution updates are available in (Wibisono, 2018). Additionally, the $\mathcal{W}_2$ distance

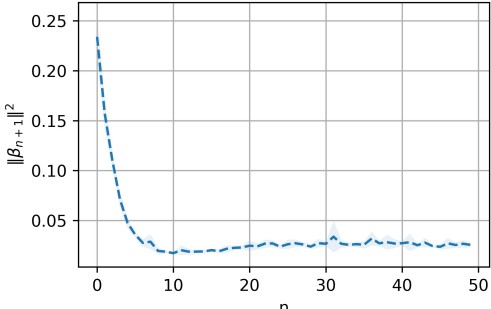

Figure 3: Behavior of inexact error $\|\hat{\beta}_{n+1}\|^2_{L_2(\rho^m_{n+1})}$ ($\tau = 0.1, \xi = 0.2$).

between two Gaussian distributions can be computed explicitly. In the experiment, we set the initialization Gaussian distribution to be $\mathcal{N}(0, 100)$, step size $\xi = 0.1$, and iterations equal to 60. Our implementation is based on the implementation of Salim et al. (2020).

In Figure 1, the distribution of particles, represented by the histogram, approximates $\rho_n$ and converges to $\rho^*$ after several iterations ($\approx 40$ iterations in this experiment). The linear convergence result of $D_{KL}(\rho_n \| \nu)$ in Figure 1d demonstrates a sharper bound holds for $\mu$-convex objective with respect to (Yao & Yang, 2022; Cheng et al., 2024), as Theorem 3.10 suggests.

## 4.2 Mean-field neural network training with entropy regularization

In practice, the optimization problem (14) typically lacks an explicit solution. Therefore, we can use particle methods to approximate the time-evolving probability distributions, and we can solve an approximate $T^{n+1}$ using functional approximation methods (cf. more details in Appendix Section D.1). In this experiment, we minimize the MFLD entropy-regularized total objective (11) with $\varphi(\theta, x) = \tanh(\theta^T x)$, which satisfies the Wasserstein PL inequality (see Corollary 3.7), but is not geodesically convex. The parameters are set as $d = 2$, $\lambda = 0.1$, $\tau = \{0.05, 0.1\}$, with the number of particles $m = 100$ and a discretized step size of $\xi = 0.2$. We generate $N = 1000$ training data samples using a teacher model $y = \sin(\alpha^T x)$, where $x \sim \mathcal{N}(0, I)$.

Our goal is to compare the proximal algorithm with the neural network-based functional approximation (54) with the noisy gradient descent algorithm (5). We first randomly generated a dataset and then conducted 5 repeated experiments for both $\tau = 0.05$ and $\tau = 0.1$ on the same generated data. For each experiment, a new weight empirical distribution is generated from the standard Gaussian distribution, and both algorithms will use the same weight as the initial value. To learn the optimal transport map, we train a RealNVP normalization flow for each step using SGD optimizer with learning rate 0.005 and 150 iterations by default unless explicitly mentioned (see Appendix D.2 for more details). In Figure 2a and Figure 2b, we observe that both the $L^2$-regularized loss $R$ and the total objective $F_\tau$ converge under two algorithms, where the nearest neighbor estimator (Kozachenko & Leonenko, 1987) is used to estimate $\int_\Theta \rho \log \rho$.

To better depict the convergence rate of the Wasserstein proximal algorithm and the Langevin algorithm, we obtain a reference $\tilde{\rho}^*_\tau$ of $\rho^*$, by running the noisy gradient descent algorithm with very small step size 0.001 and $m = 1000$ particles. In the early training phase of both algorithms, $\mathcal{W}^2_2(\rho^m_n, \tilde{\rho}^*_\tau)$ is dominated by $\mathcal{W}^2_2(\rho^m_n, \rho^*)$ and exhibits a linear convergence above the black dash-dot line as shown in Figure 2c and Figure 2d. Within this phase, the Wasserstein proximal algorithm demonstrates a faster linear rate thanks to the unbiased linear convergence nature of $\mathcal{W}^2_2(\rho^m_n, \rho^*)$ (cf. Corollary 3.7). However, both algorithms have similar biases at convergence, for which we conjecture that the particle discretization error of $\rho^m_n$ dominates $\mathcal{W}^2_2(\rho^m_n, \tilde{\rho}^*_\tau)$ while close to convergence. We validate our conjecture through further experiments in Appendix D.2.

We also explore the empirical behavior of $\|\hat{\beta}_{n+1}\|^2_{L_2(\rho^m_{n+1})}$ as $n$ increases in Figure 3. $\nabla \log \rho$ is approximately computed by a kernel approach, where we use a Gaussian kernel with a bandwidth 0.5. We observe that the approximate $\|\hat{\beta}_{n+1}\|^2_{L_2(\rho^m_{n+1})}$ is bounded, suggesting the boundedness assumption $\epsilon_n \leq \epsilon$ in Theorem 3.12.

## 5  Conclusion

In this work, we provide a convergence analysis of the Wasserstein proximal algorithm under a Wasserstein PL inequality, without assuming geodesic convexity. Our analysis also improves upon existing convergence rates when strong geodesic convexity holds. We further analyze inexact gradient variants under an additional geodesic semiconvexity condition. When applied to proximal training of mean-field neural networks, we prove linear convergence of the entropy-regularized total objective, and show empirically that proximal training converges faster than the noisy gradient descent algorithm.

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

# A Background on optimal transport and Wasserstein space

## A.1 Wasserstein distance and optimal transport

The squared 2-Wasserstein distance is defined as the solution to the Kantorovich problem

$$\mathcal{W}_2^2(\rho, \tilde{\rho}) := \min_{\pi \in \Pi(\rho, \tilde{\rho})} \int_{\Theta \times \Theta} \left\| \theta - \tilde{\theta} \right\|^2 d\pi(\theta, \tilde{\theta}), \tag{23}$$

where $\Pi(\rho, \tilde{\rho}) \subset \mathcal{P}_2(\Theta \times \Theta)$ is the set of all coupling distributions with marginals $\rho$ and $\tilde{\rho}$. The optimal solution $\pi^*$ is called the optimal transport plan. When $\rho \in \mathcal{P}_2^a(\Theta)$, it is known from celebrated Brenier's theorem that the solution $T_\rho^{\tilde{\rho}}$ of Monge's problem exists, and the optimal transport plan is $\pi^* = (\mathbf{id}, T_\rho^{\tilde{\rho}})_{\#}\rho$.

## A.2 Wasserstein subdifferential and $\mu$-convex functionals

**Definition A.1** (Frechet subdifferential, Definition 10.1.1 in Ambrosio et al. (2008)). Let $F : \mathcal{P}_2(\Theta) \to (-\infty, +\infty]$ be proper, lower semicontinuous, and let $\rho \in D(|\partial F|)$ where $|\partial F|(\rho)$ denotes the metric slope (Ambrosio et al., 2008). We say that $\mathbf{v} \in L^2(\rho; \Theta)$ belongs to the Frechet subdifferential $\partial F(\rho)$ of $F$, written as $\mathbf{v} \in \partial F(\rho)$, if

$$F(\tilde{\rho}) \geq F(\rho) + \int_\Theta \langle \mathbf{v}(\theta), T_\rho^{\tilde{\rho}}(\theta) - \theta \rangle d\rho(\theta) + o(\mathcal{W}_2(\rho, \tilde{\rho})).$$

And $\mathbf{v}$ is called strong subdifferential. Also, $\mathbf{v} \in \partial F(\rho)$ with minimal $\|\cdot\|_{L_2(\rho)}$ norm is denoted as $\partial^\circ F(\rho)$, and it is unique (cf. Lemma 10.1.5 in Ambrosio et al. (2008)).

If $\partial F(\rho)$ is not empty, then we say $F$ is subdifferentiable at $\rho$. $F$ is differentiable at $\rho$ if both $F$ and $-F$ are subdiffrentiable.

**Definition A.2** (First variation). Let $F : \mathcal{P}_2(\Theta) \to (-\infty, +\infty]$ be proper, lower semicontinuous, and $\rho \in \mathcal{P}_2(\Theta)$. The first variation $\frac{\delta F}{\delta \rho}(\rho) : \Theta \to \mathbb{R}$ is defined as

$$\frac{d}{d\varepsilon} F(\mu + \varepsilon \chi) \Big|_{\varepsilon=0} = \int \frac{\delta F}{\delta \rho}(\rho) d\chi \tag{24}$$

for any perturbation $\chi = \tilde{\rho} - \rho$ with $\tilde{\rho} \in \mathcal{P}_2(\Theta)$.

**Definition A.3** ($\mu$-convexity along geodesic, Section 10.1.1 of Ambrosio et al. (2008)). A proper, lower semicontinuous functional $F$ is said to be $\mu$-convex along geodesic ($\mu \in \mathbb{R}$) at $\rho \in \mathcal{D}(F) \cap D(|\partial F|)$, if for all $\tilde{\rho} \in \mathcal{D}(F)$,

$$F(\tilde{\rho}) \geq F(\rho) + \int_\Theta \langle \mathbf{v}(\theta), T_\rho^{\tilde{\rho}}(\theta) - \theta \rangle d\rho(\theta) + \frac{\mu}{2} \mathcal{W}_2^2(\rho, \tilde{\rho}), \tag{25}$$

where $\mathbf{v} \in \partial F(\rho)$. In particular, if $\mu \geq 0$, we call F is **geodesically convex**; If $\mu < 0$, we call $F$ is **geodesically semiconvex**. We refer to Section 9 of Ambrosio et al. (2008) for definitions of $\mu$-convexity (and semiconvexity) in Euclidean space, which are similar to the definition above.

**Definition A.4** (Weak convergence). Let $\mathcal{C}_b(\Theta)$ be the set of all continuous bounded functions on $\Theta$ and $\mathcal{M}(\Theta)$ be the set of all finite signed measures on $\Theta$. We say that $\rho_k \in \mathcal{M}(\Theta)$ converges to $\rho \in \mathcal{M}(\Theta)$ weakly if for every $\phi \in \mathcal{C}_b(\Theta)$,

$$\lim_{k \to \infty} \int_\Theta \phi d\rho_k = \int_\Theta \phi d\rho. \tag{26}$$

The weak convergence is also called narrow convergence in the literature Santambrogio (2015).

# B Technical lemmas

Recall that for a general metric space $(Z, d)$, the Hopf-Lax semigroup is defined as

$$u(z, \xi) = \inf_{z' \in Z} \left\{ f(z') + \frac{1}{2\xi} d(z', z)^2 \right\} \tag{27}$$

for $z \in Z$ such that $f(z) < +\infty$.

**Lemma B.1** (Proposition 3.1 and 3.3 in Ambrosio et al. (2014)). *Let*

$$D_+(z,\xi) := \sup \limsup_{n\to\infty} d(z, z'_n), \qquad D_-(z,\xi) := \inf \liminf_{n\to\infty} d(z, z'_n), \tag{28}$$

*where the supremum and the infimum run among all minimizing sequences $(z'_n)$. For $z \in Z$ such that $f(z) < +\infty$, we define $t^*(z) = \sup\{t > 0 : u(t,\xi) > -\infty\}$. Then if $f(z) < +\infty$, we have*

*(a) $D_+(z,\xi) = D_-(z,\xi)$ holds for $\xi \in (0, t^*(z))$ except for at most countable exceptions;*

*(b) If and only if $D_+(z,\xi) = D_-(z,\xi)$, the map $\xi \to u(z,\xi)$ is differentiable in $(0, t^*(z))$ and*

$$\frac{d}{d\xi} u(z,\xi) = -\frac{D_+^2(z,\xi)}{2\xi^2} = -\frac{D_-^2(z,\xi)}{2\xi^2}.$$

**Lemma B.2** (Existence of minimizer). *If $F : \mathcal{P}_2(\Theta) \to (-\infty, \infty]$ is weakly lower semicontinuous, then proximal algorithm (2) admits a minimizer.*

*Proof.* The proof is essentially contained in Section 10.1 of Ambrosio et al. (2008). For the sake of completeness, we provide a proof here. We only need to show $\mathcal{B}_r(\rho) = \{\nu | \mathcal{W}_2^2(\rho, \nu) < r\}$ is weakly-precompact for any fixed $r > 0$. By Prokhorov's theorem, it suffices to prove the tightness of $\mathcal{B}_r(\rho)$, i.e., there is a sequence of compact sets $(K_i)_{i\in\mathbb{N}}$ such that

$$\nu(\Theta\backslash K_i) \leq 1/i, \quad \forall \nu \in \mathcal{B}_r(\rho). \tag{29}$$

We prove for any $\varepsilon > 0$, we can find compact set $K$ such that

$$\nu(\Theta\backslash K) \leq \varepsilon, \quad \forall \nu \in \mathcal{B}_r(\rho). \tag{30}$$

by contradiction. Assume there exists $\varepsilon$, for any compact $K$, there exists $\nu \in \mathcal{B}_r(\rho)$ such that $\nu(\Theta\backslash K) > \varepsilon$. As a singleton $\{\rho\}$ constitutes a tight family, we can find a compact set $K_{\rho,\varepsilon}$ such that

$$\rho(\Theta\backslash K_{\rho,\varepsilon}) < \varepsilon/2.$$

We define a compact set $U_R = \{\tilde{\theta}| \min_{\theta \in K_{\rho,\varepsilon}} \|\theta - \tilde{\theta}\|^2 \leq R,\}$. Then there exists $\nu$ such that $\nu(\Theta\backslash U_{3r/\varepsilon}) > \varepsilon$. However, $\mathcal{W}_2^2(\rho, \nu) \geq \frac{\varepsilon}{2}\frac{3r}{\varepsilon} = \frac{3}{2}r$, contradiction. $\square$

*Remark* B.3 (Conditions on weak lower semicontinuity). We give several examples that the functional $F$ satisfies weak lower semicontinuity.

- If $f : \Theta \to (-\infty, +\infty]$ is lower semicontinuous and its negative part has a 2-growth, then $F = \int_\Theta f d\rho$ is lower semicontinuous with respect to $\mathcal{W}_2$ topology, see Example 9.3.1 in Ambrosio et al. (2008). If $f : \Theta \to (-\infty, +\infty]$ is lower semicontinuous and bounded from below, then $F = \int_\Theta f d\rho$ is weakly lower semicontinuous, see Example 9.3.1 in Ambrosio et al. (2008).

- For conditions that ensure the weak lower semicontinuity of internal energy, we refer to Section 9.3 in Ambrosio et al. (2008) for details. Specifically, $\int_\Theta \rho log \rho$ is weakly lower semicontinuous.

*Remark* B.4 (Conditions on (1) in Assumption 1). If $F : \mathcal{P}_2(\Theta) :\to (-\infty, +\infty]$ is proper, lower semicontinuous (with respect to $\mathcal{W}_2$ topology), $\mu$-convex ($\mu \in \mathbb{R}$) along generalized geodesics, and bounded from below, then the proximal algorithm (2) admits a minimizer for any $\xi > 0$. For details, we refer to Section 10.3 in Ambrosio et al. (2008), where the condition is even weaker, being defined only on $\mathcal{W}_2$-bounded sets.

**Lemma B.5** (Satisfaction of Assumption 2 on $\mathcal{P}_2(\Theta)$ with compact set $\Theta$). *Under Assumption 1, if $\Theta$ is compact, then Assumption 2 holds.*

*Proof.* By Proposition 7.17 in Santambrogio (2015), when $\Theta$ is compact, for fixed $\rho \in \mathcal{P}_2(\Theta)$, the first variation of $\mathcal{W}_2(\cdot, \rho)$ is well-defined for all $\tilde{\rho} \in \mathcal{P}_2^a(\Theta)$. Similar to Proposition 8.7 in Santambrogio (2015), by standard calculus of variation followed by gradient operation, we have

$$\nabla \frac{\delta F}{\delta \rho}(\rho_\xi) = \frac{T_{\rho_\xi}^\rho - \mathbf{id}}{\xi}. \tag{31}$$

Please refer to Lemma B.1 in Yao et al. (2024) for similar arguments. Therefore, Assumption 2 holds. $\square$

**Lemma B.6.** *Assume $u : [0, \xi] \to (0, +\infty)$ is a decreasing function, $\partial_t u(t) \leq -C(t)u(t)$ almost everywhere for $t \in [0, \xi]$, $C(t) > 0$ for every $t \in [0, \xi]$, then*

$$u(\xi) \leq u(0) \exp\left( \int_0^\xi -C(t)dt \right). \tag{32}$$

*Remark* B.7. Lemma B.6 extends the classical Gronwall lemma, which requires everywhere differentiability, to the case with almost everywhere differentiability and monotonicity.

*Proof.* By the monotonicity of $u(t)$, we construct a function $g(t) = \ln(u(t))$. It is a decreasing function and $\dfrac{dg(t)}{dt} = \dfrac{\partial_t u(t)}{u(t)} \leq -C(t)$ almost everywhere on $[0, \xi]$. By properties of the Lebesgue integral, we have

$$\int_0^\xi \frac{dg(t)}{dt}dt \leq \int_0^\xi -C(t)dt. \tag{33}$$

Since $g$ is decreasing, by Proposition 6.6 in Komornik (2016),

$$g(\xi) - g(0) \leq \int_0^\xi \frac{dg(t)}{dt}dt. \tag{34}$$

Thus,

$$u(\xi) \leq u(0) \exp\left( \int_0^\xi -C(t)dt \right). \tag{35}$$

$\square$

**Lemma B.8.** *Assume $T_{\bar{\rho}_n}^{\bar{\rho}_{n+1}}$ is $C^2$ diffeomorphism. If $\bar{\rho}_n \in C^1(\Theta)$, then $\bar{\rho}_{n+1} \in C^1(\Theta)$.*

*Proof.* The change variable formula of probability density is,

$$T_\# \rho(\theta) = \rho(T^{-1}(\theta)) \cdot |\det D(T^{-1})(\theta)|, \tag{36}$$

where $D(T^{-1})$ is the Jacobian matrix of $T^{-1}$. Since $T$ is a $\mathcal{C}^2$ diffeomorphism, then $T^{-1}$ is $\mathcal{C}^2$ mapping. Thus, $\rho \circ (T^{-1})$ is $\mathcal{C}^1(\Theta)$. Since $T^{-1}$ is diffeomorphism, then $D(T^{-1})$ is not singular and $|\det D(T^{-1})|$ is $C^1(\Theta)$. And thus $T_\# \rho$ is $\mathcal{C}^1(\Theta)$. $\square$

*Remark* B.9. We have an example from normalizing flow that can be $\mathcal{C}^2$ diffeomorphism. The coupling layer in Real NVP (Dinh et al., 2016) has the following structure,

$$T : \begin{cases} \alpha_{1:\tilde{d}} = \theta_{1:\tilde{d}}, \\ \alpha_{\tilde{d}+1:d} = \theta_{\tilde{d}+1:d} * \exp(s(\theta_{1:\tilde{d}})) + t(\theta_{1:\tilde{d}}), \end{cases} \tag{37}$$

where $*$ refers to the pointwise product. $T$ is naturally invertible (The Jacobian matrix is always non-singular) and the reverse is,

$$T^{-1} : \begin{cases} \theta_{1:\tilde{d}} = \alpha_{1:\tilde{d}}, \\ \theta_{\tilde{d}+1:d} = (\alpha_{\tilde{d}+1:d} - t(\alpha_{1:\tilde{d}})) * \exp(-s(\alpha_{1:\tilde{d}})). \end{cases} \tag{38}$$

It is not hard to see that if $s(\cdot) : \mathbb{R}^{\tilde{d}} \to \mathbb{R}^{d-\tilde{d}}$ and $t(\cdot) : \mathbb{R}^{\tilde{d}} \to \mathbb{R}^{d-\tilde{d}}$ are $\mathcal{C}^2$ maps (i.e., represented by fully connected neural network with smooth activation function), then $T$ is restricted to be $\mathcal{C}^2$ diffeomorphism.

**Lemma B.10.** *Under Assumption 3, if $\bar{\rho}_n \in C^1(\Theta)$ and $\bar{\rho}_{n+1}$ is the exact solution of the Wasserstein proximal algorithm (2), then $\beta_{n+1} = 0$.*

*Proof.* Since $\overline{\rho}_{n+1}$ is the exact solution of (2), then $\overline{\rho}_{n+1} \in D(|\partial F|)$ and $\frac{T_{\overline{\rho}_{n+1}}^{\overline{\rho}_n} - \mathbf{id}}{\xi} \in \partial F(\overline{\rho}_{n+1})$ by Lemma 10.1.2 in Ambrosio et al. (2008). As $\overline{\rho}_{n+1} \in \mathcal{C}^1(\Theta)$, then $\nabla \frac{\delta F}{\delta \rho}(\overline{\rho}_{n+1})$ is the unique strong subdifferential by Lemma 10.4.1 in Ambrosio et al. (2008). Therefore,

$$T_{\overline{\rho}_{n+1}}^{\overline{\rho}_n} - \mathbf{id} - \xi \nabla \frac{\delta F}{\delta \rho}(\overline{\rho}_{n+1}) = 0.$$

□

## C   Proofs

**Proof of Lemma 3.2.** Lemma B.2 guarantees the existence of a minimizer of the JKO scheme or proximal algorithm. Thus we can define

$$z_\xi = \arg \min_{z \in \mathcal{P}_2(\mathbb{R}^d)} u(z, \xi).$$

Then Lemma B.1 implies that $D_+(z, \xi) = D_-(z, \xi) = d(z, z_\xi)$ in $(0, t^*(z))$ except for at most countable exceptions, from which we can conclude the desired Lemma 3.2. □

*Proof of Theorem 3.5.* It suffices to prove one-step contraction. We begin with

$$
\begin{aligned}
\partial_\xi (u(\rho, \xi) - F^*) &= -\frac{\mu}{2\xi(1+\mu\xi)} \mathcal{W}_2^2(\rho_\xi, \rho) - \frac{1}{2(1+\mu\xi)\xi^2} \mathcal{W}_2^2(\rho_\xi, \rho) && \text{(by Lemma 3.2)} \\
&= -\frac{\mu}{2\xi(1+\mu\xi)} \mathcal{W}_2^2(\rho_\xi, \rho) - \frac{1}{2(1+\mu\xi)\xi^2} \int_\Theta \left\| T_{\rho_\xi}^\rho - \mathbf{id} \right\|^2 d\rho_\xi && \text{(by Brenier's theorem)} \\
&\le -\frac{\mu}{2\xi(1+\mu\xi)} \mathcal{W}_2^2(\rho_\xi, \rho) - \frac{1}{2(1+\mu\xi)} \int_\Theta \left\| \nabla \frac{\delta F}{\delta \rho}(\rho_\xi) \right\|^2 d\rho_\xi && \text{(by Assumption 2)} \\
&\le -\frac{\mu}{2\xi(1+\mu\xi)} \mathcal{W}_2^2(\rho_\xi, \rho) - \frac{\mu}{(1+\mu\xi)}(F(\rho_\xi) - F^*) && \text{(by PL (15))} \\
&= -\frac{\mu}{1+\mu\xi}(u(\rho, \xi) - F^*). && \text{(by Def 3.1)}
\end{aligned}
$$

Using Lemma B.6 to deal with the technical issue of almost everywhere differentiability, we have

$$u(\rho, \xi) - F^* \le (u(\rho, 0) - F^*) \exp\left( \int_0^\xi -\frac{\mu}{1+\mu t} dt \right) = (F(\rho) - F^*) \frac{1}{1+\mu\xi}.$$

Invoking the PL inequality (15) once again, we obtain that

$$
\begin{aligned}
(F(\rho) - F^*) \frac{1}{1+\mu\xi} &\ge u(\rho, \xi) - F^* = F(\rho_\xi) - F^* + \frac{1}{2\xi} \mathcal{W}_2^2(\rho_\xi, \rho) \\
&\ge F(\rho_\xi) - F^* + \frac{\xi}{2} \int_\Theta \left\| \nabla \frac{\delta F}{\delta \rho}(\rho_\xi) \right\|^2 d\rho_\xi \ge (1+\mu\xi)(F(\rho_\xi) - F^*).
\end{aligned}
$$

□

**Definition C.1** (Uniform log-Sobolev inequality)**.** There is a constant $\mu > 0$ such that for any $\rho \in \mathcal{P}_2(\Theta)$, its Gibbs proximal distribution $q_\rho$ defined as,

$$q_\rho(\theta) \propto \exp\left( -\frac{1}{\tau} \frac{\delta R}{\delta \rho}(\rho)(\theta) \right) \tag{39}$$

satisfies the log-Sobolev inequality (8) with the constant $\mu$.

**Proof of Corollary 3.7.** We divide our proof into four parts.

(1) We prove the satisfaction of Assumption 1 and the geodesic semiconvexity of $F_\tau$.

(2) We prove the PL inequality.

(3) We prove the satisfaction of Assumption 2 by showing that $\nabla \frac{\delta F_\tau}{\delta \rho}(\rho_\xi) = \partial^\circ F_\tau(\rho_\xi)$.

(4) With the previous three parts, we can get the linear convergence of the function value. The last part is devoted to obtaining a convergence rate of $\mathcal{W}_2$ distance using some structure of MFLD.

**Our proof is as follows,**

(1) The weak lower semicontinuity of $F_\tau$ is verified in Section 5.1 in Chizat (2022), and thus a minimizer is admitted by Lemma B.2. The geodesic semiconvexity follows from Lemma A.2 in Chizat (2022) and the proof relies on (41) below.

(2) Now we prove PL inequality. The training risk $R : \mathcal{P}_2(\Theta) \to (-\infty, +\infty]$ has linear convexity if the loss function $l$ is convex (in the Euclidean sense). By Proposition 5.1 in (Chizat, 2022), the $L^2$-regularized training risk $R$ in (4) satisfies the uniform LSI assumption (Assumption 3 in Chizat (2022)). Next, we shall show that these assumptions imply the relaxed PL-inequality defined in (15). By the entropy sandwich bound Lemma 3.4 in (Chizat, 2022) (which relies on linear convexity of $F_\tau$),

$$\tau D_{\mathrm{KL}}(\rho \| q_\rho) \geq F_\tau(\rho) - F_\tau(\rho^*). \tag{40}$$

Therefore,

$$\int_\Theta \left\| \nabla \frac{\delta F_\tau}{\delta \rho}(\rho) \right\|^2 d\rho = \int_\Theta \left\| \nabla \frac{\delta R}{\delta \rho}(\rho) + \tau \nabla \log(\rho) \right\|^2 d\rho$$
$$= \tau^2 J_{q_\rho}(\rho) \geq 2\mu_\tau \tau^2 D_{\mathrm{KL}}(\rho \| q_\rho) \geq 2\mu_\tau \tau (F_\tau(\rho) - F_\tau(\rho^*)).$$

Thus, the functional $F_\tau$ satisfies the Wasserstein PL-inequality with parameter $\tau \mu_\tau$.

(3) Under Assumption 1, $(T_{\rho_\xi}^\rho - \mathbf{id})/\xi$ is a strong subdifferential at $\rho_\xi \in \mathcal{P}_2^a(\Theta)$, and $\rho_\xi \in D(|\partial F_\tau|)$ by Lemma 10.1.2 in Ambrosio et al. (2008). To prove that Assumption 2 holds, we only need to prove that,

$$\textit{If } \rho \in D(|\partial F_\tau|) \cap \mathcal{P}_2^a(\Theta), \textit{ then } \nabla \frac{\delta F_\tau}{\delta \rho}(\rho) = \partial^\circ F_\tau(\rho).$$

Our proof for (3) below highly relies on the proof of Theorem 10.4.13 in Ambrosio et al. (2008).

**Step 1.** We first need to derive some conditions similar to equations (10.4.58) and (10.4.59) in Ambrosio et al. (2008). Under the assumptions of Corollary 3.7, $R$ satisfies the following smoothness condition with $\tilde{L} > 0$ by Proposition 5.1 in Chizat (2022). Thus, $\forall \theta, \tilde{\theta} \in \Theta$ and $\forall \rho, \tilde{\rho} \in \mathcal{P}_2(\Theta)$, we have

$$\left\| \nabla \frac{\delta R}{\delta \rho}(\rho)(\theta) - \nabla \frac{\delta R}{\delta \rho}(\tilde{\rho})(\tilde{\theta}) \right\| \leq \tilde{L}(\|\theta - \tilde{\theta}\|_2 + \mathcal{W}_2(\rho, \tilde{\rho})). \tag{41}$$

By Lemma A.2 in Chizat (2022), relying on (41), choosing $\mathbf{r} = \mathbf{id} + \mathbf{t}$ with $\mathbf{t} \in C_c^\infty(\Theta)$,

$$\lim_{t \to 0} \frac{R((\mathbf{id} + t\mathbf{t})_{\#}\rho) - R(\rho)}{t} = \int_\Theta \left\langle \nabla \frac{\delta R}{\delta \rho}(\rho), \mathbf{r} - \mathbf{id} \right\rangle d\rho, \tag{42}$$

which is the key condition similar to equations (10.4.58) and (10.4.59) in Ambrosio et al. (2008) that we want to obtain.

Furthermore, a by-product is that (42) suggests that $\nabla \frac{\delta R}{\delta \rho}(\rho)$ is a (unique) strong subdifferential and $\left\| \nabla \frac{\delta R}{\delta \rho}(\rho) \right\|_{L_2(\rho)} = |\partial R|(\rho)$ for all $\rho \in \mathcal{P}_2^a(\Theta)$, see Lemma A.2 in Chizat (2022). It is not hard to verify that $\left\| \nabla \frac{\delta R}{\delta \rho}(\rho) \right\|_{L_2(\rho)}$ is finite at any $\rho \in \mathcal{P}_2(\Theta)$ because (41) ensures 2-growth of $\left\| \nabla \frac{\delta R}{\delta \rho}(\rho)(\cdot) \right\|^2$.

In **Step 2**, we conduct a proof similar to Theorem 10.4.13 in Ambrosio et al. (2008).

**Step 2.** By Lemma 10.4.4 in Ambrosio et al. (2008), (42), and the fact that $\rho \in \mathcal{P}_2^a(\Theta) \cap D(|\partial F|)$, for $\mathbf{t} \in C_c^\infty(\Theta)$

$$- \int_\Theta \tau \rho \nabla \cdot \mathbf{t} d\theta + \int_\Theta \left\langle \nabla \frac{\delta R}{\delta \rho}(\rho), \mathbf{t} \right\rangle d\rho \geq -|\partial F_\tau(\rho)| \, \|\mathbf{t}\|_{L_2(\rho)}. \tag{43}$$

By (44), $\forall \theta, \tilde{\theta} \in \Theta, \forall \rho \in \mathcal{P}_2(\Theta)$,

$$\left\| \nabla \frac{\delta R}{\delta \rho}(\rho)(\theta) - \nabla \frac{\delta R}{\delta \rho}(\rho)(\tilde{\theta}) \right\| \leq \tilde{L} \|\theta - \tilde{\theta}\|_2. \tag{44}$$

Therefore, $\nabla \frac{\delta R}{\delta \rho}(\rho)(\cdot)$ is Lipschitz and locally bounded. Therefore, following the same argument of Theorem 10.4.13 in Ambrosio et al. (2008), we obtain that

$$\nabla \frac{\delta F_\tau}{\delta \rho}(\rho) = \tau \frac{\nabla \rho}{\rho} + \nabla \frac{\delta R}{\delta \rho}(\rho) \in L_2(\rho) \text{ and } \left\| \nabla \frac{\delta F_\tau}{\delta \rho}(\rho) \right\|_{L_2(\rho)} \leq |\partial F_\tau(\rho)|. \tag{45}$$

where we define $\frac{\nabla \rho}{\rho} = 0$ if $\rho(\theta) = 0$. Furthermore, it is straightforward to prove $\nabla \frac{\delta F_\tau}{\delta \rho}(\rho) \in \partial F_\tau(\rho).$[1] Since $\rho \in D(|\partial F_\tau|)$ and $\rho \in D(|\partial R|)$, we have $\rho \in D(|\partial H_\tau|)$ where $H_\tau(\rho) = \int_\Theta \tau \rho \log \rho$. Therefore, $\tau \frac{\nabla \rho}{\rho} \in \partial H_\tau(\rho)$ in view of Theorem 10.4.6 in Ambrosio et al. (2008). With $\nabla \frac{\delta R}{\delta \rho}(\rho) \in \partial R(\rho)$, we have proved that $\nabla \frac{\delta F_\tau}{\delta \rho}(\rho) \in \partial F_\tau(\rho)$. Thus, $\nabla \frac{\delta F_\tau}{\delta \rho}(\rho) = \partial^\circ F_\tau(\rho)$.

(4) With the previous three parts, we can get the first inequality in Corollary 3.7 with Theorem 3.5. Next, we prove the second inequality in Corollary 3.7. Using Lemma 3.4 in (Chizat, 2022) once again, we get

$$\tau D_{\mathrm{KL}}(\rho\|\rho^*) \leq F_\tau(\rho) - F_\tau(\rho^*) \leq \tau D_{\mathrm{KL}}(\rho\|q_\rho).$$

Since $\rho^*$ also satisfies $\mu_\tau$-LSI, by Talagrand's inequality, we have

$$\mathcal{W}_2^2(\rho^*, \rho_n) \leq \frac{2}{\mu_\tau} D_{\mathrm{KL}}(\rho_n\|\rho^*) \leq \frac{2}{\mu_\tau \tau}(F_\tau(\rho_n) - F_\tau(\rho^*)) \leq \frac{2}{\mu_\tau \tau}(F_\tau(\rho_0) - F_\tau(\rho^*)) \left(\frac{1}{1 + \mu_\tau \tau \xi}\right)^{2n}. \tag{46}$$

$\square$

**Proof of Corollary 3.8.** Similar to proof of Corollary 3.7, we divide the proof into four parts.

(1) $\int_\Theta f d\rho$ is semiconvex along geodesics and $\int_\Theta \log \rho d\rho$ is convex along geodesics, see Section 9.3 of Ambrosio et al. (2008). Thus, $F$ is semiconvex along geodesics. Since $f$ is semiconvex, its negative part has 2-growth. Therefore, $\int f d\rho$ is lower semicontinuous with respect to the $\mathcal{W}_2$ topology by Remark B.3. Thus, $F$ is lower semicontinuous with respect to $\mathcal{W}_2$ topology. In addition, $F$ is bounded from below because of the LSI condition. By Remark B.4, $prox_{F,\xi}$ admits one minimizer in $\mathcal{P}_2(\Theta)$. In addition, $D(F) \subset \mathcal{P}_2^a(\Theta)$. Thus, Assumption 1 is satisfied.

(2) The PL inequality directly follows from the LSI condition.

(3) We prove that Assumption 2 is satisfied by showing $\nabla \frac{\delta F}{\delta \rho}(\rho_\xi) = \partial^\circ F(\rho_\xi)$. Theorem 10.4.13 in Ambrosio et al. (2008) already incorporates KL divergence as a special case: Assume $f$ is semiconvex and lower semicontinuous, and $\{\theta | V(\theta) < \infty\}$ is not empty. Then for $F(\rho) = \int_\Theta f d\rho + \int_\Theta \rho \log \rho$ (here we assume the interaction energy to be 0), if $\rho \in D(|\partial F|) \cap \mathcal{P}_2^a(\Theta)$, then $\nabla \frac{\delta F}{\delta \rho}(\rho) = \nabla V + \nabla \log(\rho) = \partial^\circ F(\rho)$, where we assume $\nabla \log(\rho)(\theta) = 0$ if $\rho(\theta) = 0$. With Assumption 1 to guarantee $\rho_\xi \in \mathcal{P}_2^a(\Theta)$ and Lemma 10.1.2 in Ambrosio et al. (2008) to guarantee that $\rho_\xi \in D(|\partial F|)$, Assumption 2 holds.

(4) The convergence rate on $\mathcal{W}_2$ distance follows Talagrand's inequality. $\square$

---

[1] Here the proof is slightly different from Theorem 10.4.13 in Ambrosio et al. (2008) for simplicity, as we already have the finite slope property of $R$.

**Proof of Corollary 3.9.** (1) We first verify Assumption 1. By definition, $D(F) = \mathcal{P}_2^a(\mathbb{R})$ since $\int |\theta| d\rho(\theta) \leq \left(\int \theta^2 d\rho(\theta)\right)^{1/2} < \infty$. Fix $\rho \in D(F)$, $\xi > 0$, and let $x = m(\rho)$ and

$$H(m) = m + c \sin m$$

For any $\tilde{\rho} \in D(F)$, denote $\tilde{x} = m(\tilde{\rho})$. By Jensen's inequality, $W_2^2(\tilde{\rho}, \rho) \geq |m(\tilde{\rho}) - m(\rho)|^2 = |\tilde{x} - x|^2$. Conversely, for every $\tilde{x} \in \mathbb{R}$, define

$$\tilde{\rho}_{\tilde{x}} = (\mathrm{id} + \tilde{x} - x)_{\#}\rho.$$

Since $\rho \in \mathcal{P}_2^a(\mathbb{R})$, we have $\tilde{\rho}_{\tilde{x}} \in \mathcal{P}_2^a(\mathbb{R})$. Moreover, $m(\tilde{\rho}_{\tilde{x}}) = \tilde{x}$, $W_2^2(\tilde{\rho}_{\tilde{x}}, \rho) = |\tilde{x} - x|^2$. Therefore,

$$\inf_{\tilde{\rho} \in \mathcal{P}_2(\mathbb{R})} \left\{ F(\tilde{\rho}) + \frac{1}{2\xi} W_2^2(\tilde{\rho}, \rho) \right\} = \inf_{\tilde{x} \in \mathbb{R}} \left\{ H(\tilde{x})^2 + \frac{1}{2\xi} |\tilde{x} - x|^2 \right\}$$

The RHS is continuous, and coercive, i.e., $H(\tilde{x})^2 + 1/2\xi |\tilde{x} - x|^2 \to \infty$ when $|\tilde{x}| \to \infty$, hence admits a minimizer $x_\xi$. Therefore, $\rho_\xi = (\mathrm{id} + x_\xi - x)_{\#}\rho$ is a proximal minimizer and belongs to $D(F)$. Thus Assumption 1 is satisfied.

(2) We next verify the Wasserstein PL inequality. On $D(F)$, we have

$$F(\rho) = H(m(\rho))^2.$$

Thus,

$$\nabla \frac{\delta F}{\delta \rho}(\rho)(\theta) = 2H(m(\rho))H'(m(\rho)).$$

Since $H'(m) = 1 + c \cos m \geq 1 - c > 0$, we obtain

$$\int_{\mathbb{R}} \left\| \nabla \frac{\delta F}{\delta \rho}(\rho)(\theta) \right\|^2 d\rho(\theta) = 4H(m(\rho))^2 H'(m(\rho))^2 \geq 4(1 - c)^2 F(\rho).$$

Moreover, $F^* = 0$, for example by taking any $\rho \in \mathcal{P}_2^a(\mathbb{R})$ with $m(\rho) = 0$. Therefore,

$$\int_{\mathbb{R}} \left\| \nabla \frac{\delta F}{\delta \rho}(\rho)(\theta) \right\|^2 d\rho(\theta) \geq 2\mu \big( F(\rho) - F^* \big), \qquad \mu = 2(1 - c)^2.$$

Thus the Wasserstein PL inequality is satisfied.

(3) We now verify Assumption 2. It is sufficient to show that $\partial^\circ F(\rho) = \left\{ \nabla \frac{\delta F}{\delta \rho}(\rho) \right\}, \forall \rho \in D(F)$ in which case Assumption 2 holds with equality. Let $v \in \partial^\circ F(\rho)$, for any $\varphi \in C_c^\infty(\mathbb{R})$, consider the perturbation

$$\rho_t = (\mathrm{id} + t\varphi)_{\#}\rho.$$

For $t > 0$ sufficiently small, $\mathrm{id} + t\varphi$ is an increasing rearrangement, and hence it is the optimal transport map from $\rho$ to $\rho_t$. By the definition of the strong subdifferential,

$$F(\rho_t) - F(\rho) \geq t \int_{\mathbb{R}} v(\theta)\varphi(\theta) \, d\rho(\theta) + o(\mathcal{W}_2(\rho, \rho_t)) = t \int_{\mathbb{R}} v(\theta)\varphi(\theta) \, d\rho(\theta) + o(t\|\varphi\|_{L_2(\rho)})$$

On the other hand,

$$m(\rho_t) = \int_{\mathbb{R}} (\theta + t\varphi(\theta)) \, d\rho(\theta) = m(\rho) + t \int_{\mathbb{R}} \varphi(\theta) \, d\rho(\theta).$$

Thus

$$F(\rho_t) - F(\rho) = 2H(m(\rho))H'(m(\rho))t \int_{\mathbb{R}} \varphi(\theta) \, d\rho(\theta) + o(t\|\varphi\|_{L_2(\rho)}).$$

Dividing by $t > 0$ and letting $t \downarrow 0$, and then repeating the argument with $-\varphi$, gives

$$\int_{\mathbb{R}} v(\theta)\varphi(\theta) \, d\rho(\theta) = \int_{\mathbb{R}} 2H(m(\rho))H'(m(\rho))\varphi(\theta) \, d\rho(\theta).$$

Since $C_c^\infty(\mathbb{R})$ is dense in $L^2(\rho)$, we obtain

$$v(\theta) = 2H(m(\rho))H'(m(\rho)) \qquad \rho\text{-a.e.}$$

Thus Assumption 2 holds with equality.

(4) Finally, we show that $F$ is neither linearly convex nor geodesically convex when $c = 1/2$. Take two measures $\rho_0 = \mathcal{N}(\pi/2, 1), \rho_1 = \mathcal{N}(3\pi/2, 1)$. and thus we have $m(\rho_0) = \frac{\pi}{2}, \quad m(\rho_1) = \frac{3\pi}{2}$. Then for the linear interpolation $\rho_t = (1-t)\rho_0 + t\rho_1$, we have $m(\rho_t) = \frac{\pi}{2} + \pi t$. Let $G(m) = H^2(m) = (m + \frac{1}{2}\sin m)^2$. Then

$$F(\rho_t) = G\left(\frac{\pi}{2} + \pi t\right).$$

A direct computation shows $G''(m) = 2(1 + \frac{1}{2}\cos m)^2 - \sin m(m + \frac{1}{2}\sin m)$ and $G''\left(\frac{\pi}{2}\right) < 0$, so $t \mapsto F(\rho_t)$ is not convex. Hence $F$ is not linearly convex. Since the optimal transport map between $\rho_0$ and $\rho_1$ is nothing but translation, Wasserstein geodesics between $\rho_0$ and $\rho_1$ preserves the mean interpolation: $m(\tilde{\rho}_t) = (1-t)m(\rho_0) + tm(\rho_1) = \frac{\pi}{2} + \pi t$, where $\tilde{\rho}_t$ is the push-forward measure of $\rho_0$ by $(1-t)Id + tT$ with optimal map $T(\theta) = \theta + \pi$. Thus along the $W_2$ geodesic, $F(\tilde{\rho}_t) = G\left(\frac{\pi}{2} + \pi t\right)$, is also not geodesically convex.

$\square$

**Proof of Theorem 3.10.** In this proof, we will not invoke Assumption 2.
**Step 1.** We want to prove that $\mu$-geodesic convex implies, for any fixed $\rho$ and for any $\xi$,

$$F(\rho^*) \geq F(\rho_\xi) - \frac{1}{2\mu\xi^2}\mathcal{W}_2^2(\rho_\xi, \rho). \tag{47}$$

Note $\dfrac{T_{\rho_\xi}^\rho - I}{\xi}$ is a strong subdifferential at $\rho_\xi$,

$$F(\tilde{\rho}) \geq F(\rho_\xi) + \int_\Theta \left\langle \frac{T_{\rho_\xi}^\rho(\theta) - \theta}{\xi}, T_{\rho_\xi}^{\tilde{\rho}}(\theta) - \theta \right\rangle d\rho_\xi(\theta) + \frac{\mu}{2}\mathcal{W}_2^2(\rho_\xi, \tilde{\rho})$$

$$= F(\rho_\xi) + \int_\Theta \left( \left\langle \frac{T_{\rho_\xi}^\rho(\theta) - (\theta)}{\xi}, T_{\rho_\xi}^{\tilde{\rho}}(\theta) - \theta \right\rangle + \frac{\mu}{2}\left\| T_{\rho_\xi}^{\tilde{\rho}}(\theta) - \theta \right\|^2 \right) d\rho_\xi(\theta).$$

Then, we minimize both sides of (25) with respect to $\tilde{\rho} \in \mathcal{P}_2^a(\Theta)$. Clearly, $\rho^*$ minimizes the left side. For the integral term on the right-hand side, we define $T_{\rho_\xi}^{\tilde{\rho}}$ in the following,

$$T_{\rho_\xi}^{\tilde{\rho}}(\theta) - \theta = -\frac{1}{\mu\xi}\left(T_{\rho_\xi}^\rho(\theta) - \theta\right), \quad \rho_\xi\text{-a.e.} \tag{48}$$

then it minimizes the integral as it minimizes the term inside the integral almost everywhere. Therefore,

$$F(\rho^*) \geq F(\rho_\xi) - \frac{1}{2\mu\xi^2}\int_\Theta \left\| T_{\rho_\xi}^\rho(\theta) - \theta \right\|^2 d\rho_\xi(\theta) = F(\rho_\xi) - \frac{1}{2\mu\xi^2}W_2^2(\rho_\xi, \rho). \tag{49}$$

**Step 2.**

$$\begin{aligned}
\partial_\xi(u(\rho, \xi) - F^*) &= -\frac{\mu}{2\xi(1 + \mu\xi)}\mathcal{W}_2^2(\rho_\xi, \rho) - \frac{1}{2(1 + \mu\xi)\xi^2}\mathcal{W}_2^2(\rho_\xi, \rho) && \text{(by (13))} \\
&\leq -\frac{\mu}{2\xi(1 + \mu\xi)}\mathcal{W}_2^2(\rho_\xi, \rho) - \frac{\mu}{(1 + \mu\xi)}(F(\rho_\xi) - F^*) && \text{(by (47) in Step 1)} \\
&= -\frac{\mu}{1 + \mu\xi}(u(\rho, \xi) - F^*). && \text{(by Def 3.1)}
\end{aligned}$$

Using Lemma B.6 to deal with the technical issue of almost everywhere differentiability, we have

$$u(\rho, \xi) - F^* \leq (u(\rho, 0) - F^*)\exp\left(\int_0^\xi -\frac{\mu}{1 + \mu t}dt\right) = (F(\rho) - F^*)\frac{1}{1 + \mu\xi}.$$

Invoking (47) once again, we obtain that

$$(F(\rho) - F^*)\frac{1}{1 + \mu\xi} \geq u(\rho, \xi) - F^* = F(\rho_\xi) - F^* + \frac{1}{2\xi}\mathcal{W}_2^2(\rho_\xi, \rho) \geq (1 + \mu\xi)(F(\rho_\xi) - F^*).$$

**Step 3.** We derive a bound for $\mathcal{W}_2(\rho^*, \rho_n)$ here. Since for any $\rho \in \mathcal{D}(F)$,

$$F(\rho) - F(\rho^*) \geq \int_\Theta \langle \mathbf{0}, T_{\rho^*}^\rho \rangle d\rho^*(\theta).$$

Therefore, $\mathbf{0}$ is a strong subdifferential by definition of geodesic convexity. Thus, we have

$$F(\rho) - F(\rho^*) \geq \int_\Theta \langle \mathbf{0}, T_{\rho^*}^\rho \rangle d\rho^*(\theta) + \frac{\mu}{2}\mathcal{W}_2^2(\rho^*, \rho) = \frac{\mu}{2}\mathcal{W}_2^2(\rho^*, \rho) \tag{50}$$

by definition of $\mu$-convexity along geodesics. Therefore,

$$\mathcal{W}_2(\rho^*, \rho_n) \leq \sqrt{\frac{2}{\mu}F(\rho_0) - F(\rho^*)}\frac{1}{(1 + \mu\xi)^n}.$$

$\square$

**Proof of Theorem 3.12.** Under Assumption 3, since we assume $\overline{\rho}_{n+1} \in D(\partial F)$, then $\nabla\frac{\delta F}{\delta \rho}(\overline{\rho}_{n+1})$ is the unique strong subdifferential. As $F$ is $(-L)$ geodesically convex, $F$ is $(-1/\xi)$ geodesically convex,

$$F(\overline{\rho}_{n+1}) - F(\overline{\rho}_n) \leq -\xi\int_\Theta \left\langle \nabla\frac{\delta F}{\delta \rho}(\overline{\rho}_{n+1}), \frac{T_{\overline{\rho}_{n+1}}^{\overline{\rho}_n} - \mathbf{id}}{\xi} \right\rangle d\overline{\rho}_{n+1} + \frac{1}{2\xi}\mathcal{W}_2^2(\overline{\rho}_{n+1}, \overline{\rho}_n)$$

$$= -\xi\int_\Theta \left\langle \nabla\frac{\delta F}{\delta \rho}(\overline{\rho}_{n+1}), \nabla\frac{\delta F}{\delta \rho}(\overline{\rho}_{n+1}) + \beta_{n+1} \right\rangle d\overline{\rho}_{n+1}$$

$$+ \frac{1}{2\xi}\int_\Theta \left\| \xi\nabla\frac{\delta F}{\delta \rho}(\overline{\rho}_{n+1}) + \xi\beta_{n+1} \right\|^2 d\overline{\rho}_{n+1} \qquad \text{(by (18))}$$

$$= -\frac{\xi}{2}\int_\Theta \left\| \nabla\frac{\delta F}{\delta \rho}(\overline{\rho}_{n+1}) \right\|^2 d\overline{\rho}_{n+1} + \frac{\xi}{2}\int_\Theta \|\beta_{n+1}\|^2 d\overline{\rho}_{n+1}$$

$$\leq -\mu\xi(F(\overline{\rho}_{n+1}) - F(\rho^*)) + \frac{\xi}{2}\epsilon_{n+1}. \qquad \text{(by PL (15))}$$

Thus, we have the one-step contraction with an error term,

$$(1 + \mu\xi)(F(\overline{\rho}_{n+1}) - F^*) \leq (F(\overline{\rho}_n) - F^*) + \frac{\xi}{2}\epsilon_{n+1}. \tag{51}$$

Assume $\epsilon_n \leq \epsilon$ for every $n$,

$$F(\overline{\rho}_n) - F(\rho^*) \leq \frac{1}{(1 + \mu\xi)^n}(F(\rho_0) - F(\rho^*)) + \frac{1 - \frac{1}{(1+\mu\xi)^n}}{1 - \frac{1}{1+\mu\xi}}\frac{\xi\epsilon}{2}$$

$$\leq \frac{1}{(1 + \mu\xi)^n}(F(\rho_0) - F(\rho^*)) + \frac{\epsilon(1 + \mu\xi)}{2\mu}.$$

$\square$

**Proof of Corollary 3.13.** Set $A = \frac{1}{1+\mu\xi}$ and $B = \frac{\xi/2}{1+\mu\xi}$. By Eqn (51) and (Yao et al., 2024, Lemma 14), we have the following two cases.

(a) If $\epsilon_n \leq C_{exp}\gamma^n$ with $\gamma \in (0, 1)$,

$$F(\overline{\rho}_n) - F^* \leq A^n(F(\rho_0) - F^*) + \frac{BC_{exp}}{|A - \gamma|}\max\{A, \gamma\}^{n+1}. \tag{52}$$

(b) If $\epsilon_n \leq C_{poly} n^{-\zeta}$ with $\zeta > 0$, there is a constant $C(\zeta, A)$,

$$F(\overline{\rho}_n) - F^* \leq A^n(F(\rho_0) - F^*) + \frac{BC(\zeta, A)\zeta}{n^\zeta}. \tag{53}$$

$\square$

## D  Further numerical experiments and discussions

### D.1  Particle methods and functional approximation

When applying particle methods to our entropy-regularized total objective of mean-field neural network (11), the functional approximation objective (14) can be expressed as,

$$
\begin{aligned}
T^{n+1} = \arg\min_T \ \Big\{ &\frac{1}{N} \sum_{i=1}^{N} l\big(\frac{1}{m} \sum_{j=1}^{m} \varphi(T(\theta_j^n), x_i), y_i\big) + \frac{\lambda}{m} \sum_{j=1}^{m} \left\| T(\theta_j^n) \right\|^2 \\
&- \frac{\tau}{m} \sum_{j=1}^{m} \log |\det \nabla T(\theta_j^n)| + \frac{1}{2m\xi} \sum_{j=1}^{m} \left\| T(\theta_j^n) - \theta_j^n \right\|^2 \Big\},
\end{aligned}
\tag{54}
$$

where the change of variable for entropy (Mokrov et al., 2021) is utilized. In our work, we specifically employ a shallow neural network to learn the optimal transport map $T^{n+1}$ at each iteration, using the right-hand side of (54) as the loss function.

### D.2  Further discussions on MFLD experiments

**Further details of experiments.** The normalization flow has two coupling blocks as defined in Remark B.9. Both $s()$ and $t()$ are one-hidden-layer fully connected neural networks with 100 hidden dimensions and $softplus()$ activation. The two coupling layers are composed in an alternating pattern, such that the components that are left unchanged in one coupling layer are updated in the next one. For the experiment presented in Figure 3, we extend the number of training iterations for learning the OT map to 300 to promote more stable convergence of the learned OT map.

**Particle discretization error.** In this experiment, we set $\tau = 0.1$. Other settings are the same as Section 4.2. As shown in Figure 4, while $m$ increases, $\mathcal{W}_2(\rho_n^m, \tilde{\rho}_\tau^*)$ at convergence decreases for both algorithms. This experiment supports our conjecture that particle discretization error dominates the bias, if we assume $\tilde{\rho}_\tau^*$ is a good approximation of $\rho^*$.

The particle discretization error is well studied for noisy gradient descent and a quantitative uniform-in-time propagation of chaos result for the MFLD has been established in (Chen et al., 2022a). Specifically, the "distance" between the finite particle dynamics and the mean-field dynamics converges at rate $O(1/m)$ for all $t > 0$ under the uniform LSI condition. Note that even though the "distance" in theoretical analysis of uniform propagation chaos is not simply defined to be the $\mathcal{W}_2$ distance between empirical measure of finite particle dynamics and absolutely continuous measure of mean-field dynamics, it is empirically observed that the $\mathcal{W}_2$ distance also demonstrates similar propagation-of-chaos property in Figure 4b.

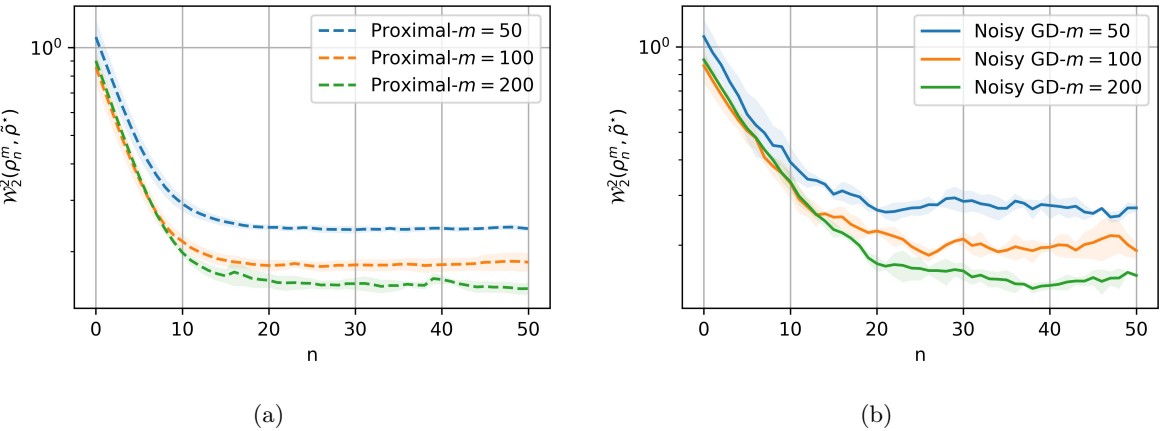

(a)                                                                (b)

Figure 4: Particle discretization error with different number of particles. We follow all the experiment settings in Section 4.2.

