# OpenReview forum: "Convergence Analysis of Wasserstein Proximal Algorithm beyond Geodesic Convexity"
_TMLR — Under review for TMLR_

### Review · Reviewer_Qnf8 · 2026-06-15

**Summary Of Contributions:**

This paper studies the convergence of the Wasserstein proximal algorithm / JKO scheme for minimizing functionals over probability measures, with a particular motivation from mean-field Langevin dynamics and training two-layer neural networks in the mean-field regime. The main contribution is a convergence analysis beyond the standard geodesic convexity assumption. Under a Wasserstein Polyak-Łojasiewicz-type inequality and a minimum-selection condition on the proximal trajectory, the authors prove a dimension-free linear convergence rate for the exact proximal algorithm. They further extend the analysis to inexact proximal steps under an additional geodesic semiconvexity assumption.

Overall, I enjoyed reading the paper. The exposition is clear and largely self-contained, the motivation is well stated, and the proofs are cleanly organized. The main theorem, Theorem 3.5, is simple and elegant, and I view it as a useful extension of the existing literature on Wasserstein proximal algorithms beyond geodesic convexity.

**Audience:**

Yes

**Audience Explanation:**

The most relevant audience is likely researchers working on optimization in the space of probability measures, Wasserstein gradient flows, sampling algorithms, variational inference, and mean-field analysis of neural networks.

**Broader Impact Concerns:**

none.

**Claims And Evidence:**

Yes

**Claims Explanation:**

Strengths
------------
The paper addresses an important and timely question: whether the stability and unbiased convergence benefits of Wasserstein proximal / JKO schemes can be obtained beyond geodesically convex objectives. The main result gives a clean linear convergence guarantee under a Wasserstein PL inequality, which is a natural analogue of the Euclidean PL condition and is well motivated by log-Sobolev-type assumptions in Langevin and mean-field Langevin settings.

The analysis is technically neat. In particular, the use of the Hopf-Lax / Moreau-Yoshida formulation to obtain the one-step contraction is elegant, and the paper clearly separates the exact and inexact proximal analyses. I also appreciate the discussion of why Wasserstein differentiability assumptions cannot simply be imposed everywhere, especially in the presence of entropy terms.

Another nice contribution is Theorem 3.10, which gives a sharper convergence rate for strongly geodesically convex objectives than previously available bounds. The Gaussian KL example and the corresponding numerical experiment provide a useful illustration of this improved rate. If the authors intend this result to be sharp in a formal sense, it would be helpful to make this explicit and state whether there is a matching lower-bound or tight example.

The numerical section is helpful in illustrating the potential advantage of proximal training over noisy gradient descent in the mean-field neural network example. The experiments are not the main contribution, but they support the theoretical message and make the proposed algorithmic direction more concrete.

------------------------------------------------
Weaknesses
--------------
My main question concerns the numerical implementation of each JKO/proximal step. The paper notes that computing the next iterate can be reformulated as an optimization problem over transport maps, and in the experiments this is approximated by particle methods and a normalizing-flow parameterization of the transport map. However, I found that the practical solvability of each proximal subproblem is not fully discussed.

In particular, it would be useful if the authors could clarify the following points.

1. How difficult is it, in practice, to solve each proximal step to the accuracy needed by the inexact theory? The inexact theorem controls the effect of the error term, but it is not entirely clear how this error relates to the optimization error incurred when training the transport map parameterization.

2. Are there any guarantees, diagnostics, or stopping criteria for the inner optimization problem used to approximate the proximal map? Since each outer iteration requires solving a nontrivial functional optimization problem, the computational cost and accuracy of this inner solver seem important for assessing the practical advantages over noisy gradient descent.

3. In the numerical experiments, the proximal method appears to converge faster in outer iterations, but it would be helpful to discuss the comparison in terms of total computational cost, since each proximal step involves training an additional transport map.

**Requested Changes:**

Besides the discussion on the solvability of each proximal subproblem mentioned above. There are a few minor comments:

1. Page 5: “weak lower semicontiunity” should be “weak lower semicontinuity”.
2. Page 8, Corollary 3.9: “Assumptions 1, 2,and” should be “Assumptions 1, 2, and”.
3. Page 15: “subdiffrentiable” should be “subdifferentiable”.
4. There are a few places where “numeric error” might read more naturally as “numerical error”.

---

> ### Author Response · Authors · 2026-06-27
>
> Thank you very much for your overall positive feedbacks! Below, we provide some clarification to your main questions.
>
> + **(On the inexact error)** We agree that the current theory does not provide a direct characterization of how accurately the transport-map optimization problem must be solved in practice to achieve a prescribed inexact error level.
>
>   Our current assumptions for inexact error are at the population level, and building the connection between this inexact error and optimization error is very challenging for proximal algorithms. This is in sharp contrast with the forward noisy GD, where each particle follows an explicit update. In such a case, one can couple the finite-particle system with its mean-field limit and obtain quantitative bounds on the discrepancy between the empirical particle law and the mean-field distribution, by the uniform propogation of chaos [CRW25]. Therefore, it provides a direct link between finite-particle and population-level dynamics.
>
>   In our proximal method, each step is defined implicitly as the solution of a Wasserstein proximal subproblem. The finite-particle algorithm first solves an **empirical**, parameterized transport-map optimization problem and then pushes forward the particle distribution by the learned map. Hence, the discrepancy from the population dynamics depends not only on particle approximation, but also on the stability of the proximal argmin with respect to empirical perturbations, the approximation power of the map class, and the numerical optimization error. Thus, it remains unclear how to obtain an analogous analysis for our implicit Wasserstein proximal scheme.
>
> + **(On computational cost)** The proximal algorithm incurs approximately 200× the runtime of noisy GD per iteration due to the inner Wasserstein optimization. However, it offers two key advantages.
>
>   + First, the proximal algorithm tolerates significantly larger step sizes while ensuring stable convergence, whereas noisy GD is more sensitive to the choice of $\xi$. Figure 4 in Appendix D.2 demonstrates that the convergence of the proximal algorithm is more stable than SGD.
>
>   + Second, our method learns optimal transport maps that can be saved and reused to abitrary number of particles without retraining. In contrast, noisy GD must be rerun from scratch whenever the number of particles changes.
>
> + **(Stopping criteria for the inner optimization problem)** A straightforward stopping criterion would be to monitor the empirical loss function given in Eq. (54). The inner optimization can be terminated when the relative change in the loss drops below a predefined tolerance threshold. However, it is still a challenging problem to assess the computational cost in theory, when finite particle error is hard to analyze, and its interplay with the inner optimization design (e.g., stopping criteria) remains highly non-trivial. In particular, the convergence of training each optimal transport map still depends on multiple factors, including the dimension $d$ and the number of particles $N$.
>
> + We will collectively correct the typos mentioned in Requested Changes. Thank you!
>
> [CRW25] Chen F, Ren Z, Wang S. Uniform-in-time propagation of chaos for mean field Langevin dynamics[C] *Annales de l'Institut Henri Poincare (B) Probabilites et statistiques. Institut Henri Poincaré*, 2025, 61(4): 2357-2404.

---

> > ### Comment · Reviewer_Qnf8 · 2026-07-14
> > **Reply to the answer**
> >
> > Thank you for the detailed response. I am satisfied with the authors’ answers and support the publication of the revised manuscript.

---

### Review · Reviewer_JUsE · 2026-06-29

**Summary Of Contributions:**

The paper analyzes the proximal point algorithm for unconstrained minimization of a functional over the Wasserstein space, and establishes linear convergence beyond the geodesically convex setting. In particular, its contributions are:
1. Linear convergence under a Wasserstein analogue of the Polyak-Lojasiewicz (PL) condition (Theorem 3.5),
2. An improved linear rate for the strongly geodesically convex case (Theorem 3.10),
3. A convergence analysis of the inexact proximal algorithm under geodesic semiconvexity (Theorem 3.12).

The framework is specialized to mean-field Langevin dynamics and KL minimization,
and is illustrated on a 1-D KL example and a small mean-field two-layer network.

**Additional Comments:**

No additional comments.

**Audience:**

Yes

**Audience Explanation:**

Yes. Convergence of proximal schemes in the Wasserstein space is an active research topic at the intersection of optimization, sampling, and the theory of mean-field neural networks, all of which are well represented in TMLR's audience.

**Broader Impact Concerns:**

None.

**Claims And Evidence:**

Yes

**Claims Explanation:**

Partially. The proofs are to the best of my knowledge correct, but I have various concerns on higher-level points and the assumptions; see comments below.

**Requested Changes:**

**Major - critical:**
1. Solving a proximal optimization problem is itself a challenge, and it is not clear to me that solving the proximal optimization problem is easier than solving the original problem. In particular, the strategy proposed in Remark 3.3 could be used directly to optimize the functional $F$. The authors should carefully explain why the proximal optimization problem is easier to solve than the original problem.
2. On this note, the computational complexity associated with the optimization problem in the proximal algorithm is far from being dimension-free. As a consequence, I find the emphasis on dimension-free misleading.
3. On a similar note, in the numerical results, be more upfront with the computational cost associated with their method. In particular, what is the difference in computational cost between the methods in Figure 2? In general, the number of iterations is not necessarily a good proxy for the computational cost.
4. The numerical examples are toy examples and merely verify the theory, but do not provide additional insights (e.g., on the effect of the parameters). I personally do not consider this to be a strict requirement, but it is definitely a missed opportunity to provide the readers with more practical insights. Also, strictly speaking, the theory demands absolute continuity of all measures, while all numerical examples use particles—acceptable, but should be acknowledged in the paper.
5. Assumption 2 should be a lemma more than an assumption: The assumption is essentially an optimality condition for the proximal optimization problem; the authors prove that this assumption is automatically satisfied in the compact case, but the authors, on the first page, claim to be interested in the non-compact case. My understanding is that necessary conditions for optimality in the Wasserstein space would yield that condition.

**Major - would strengthen:**
1. Assumption 1(2) feels somewhat superfluous: The authors situate themselves in the space of absolutely continuous measures. Why is that the case? The theory of Wasserstein gradient flows exists more broadly, even for measures that are not absolutely continuous (such as the empirical distributions used later in the numerical examples).
2. While valid, I do not find Corollary 3.9 very convincing: The problem is unconstrained and the cost only depends on the mean, which effectively immediately reduces the problem to a problem over the real line. A more practical example would strengthen the paper.


**Minor**
1. The authors say that the convergence rate obtained via PL, when specialized to the geodesically convex case, provides an improved convergence rate. However, the proof of this result (Theorem 3.10) is not a simple corollary but uses a different analysis. My impression is that the authors address both problem settings and not that only one follows from the other.
2. Position with respect to the literature is slightly imprecise. E.g., The paper says that “However, convergence analysis for non-geodesically convex objective functionals is missing.”, but this statement does not seem to be true; e.g., see the paper “Non-geodesically-convex optimization in the Wasserstein space” and there is a recent preprint on inexact JKO (https://arxiv.org/pdf/2505.23517), which is however not cited in the context of the inexact update. Please improve the literature review, stressing the differences and novelty compared to the existing works.
3. Can you explain more in detail the sentence “more precisely” at the beginning of page 7?
4. How does $K$ and the supremum of the gradient of \varphi affect the quality of the bounds in (17)?
5. In Section 4.2, the authors claim the Wasserstein PL condition is satisfied and refer the reader to Corollary 3.7… which however shows something different. I suggest stating that the functional satisfies the PL condition, so that the reader does not have to go through the proofs.
6. Is Lemma B6 novel? It appears to be a very standard result—no problem if it is, but it should be indicated more clearly.
7. Proofs:
   - I suggest saying that the chain of inequalities holds for ae $\xi$, but that this is enough to apply Lemma B6.
   - The proof of Theorem 3.10 applies when $\Theta$ is $\mathbb R^d$, and not of a general Theta, right?
   - In the proof of Theorem 3.10, you should prove that the map $T_{\rho\xi}^{\rho}$ is optimal; else, the equality in (49) does not hold.
8. Other minor comments:
   - I suggest using \text or \mathrm for symbols like prox and exp in Corollary 3.13.
   - The paper contains many typos: extra parenthesis right before 3.2, in Section A2, missing capitalization in (4) on page 22, just to name a few. Please proofread the paper more carefully.
   - I suggest converting Corollary 3.9 to an example.
   - Proof of Lemma 3.2 at the beginning of appendix C should be italic and not bold.
   - Use large parenthesis in expressions like those in (5).
   - The reference style is often wrong, e.g., Chapter 8 of Santambrogio instead of Chapter 8 of (Santambrogio).
   - The abbreviation Eqn on page 23 is informal.
   - Please only number the equations you later reference.
   - Please unify the style of the references (e.g., some have doi, others do not).
   - I suggest improving the quality of the plots (e.g., larger ticks and labels).

---

### Review · Reviewer_yKGb · 2026-07-17

**Summary Of Contributions:**

This paper considers the problem of minimizing functionals of measure which is an abstract problem that arises in practice in many machine learning applications.

The problem is as follows.
Let $\Theta \subseteq \mathbb R^d$.
Given some $F:\mathcal P_2(\Theta)\to \mathbb R$ the task is to find $\min_{\rho \in \mathcal P_2(\Theta)} F(\rho)$ and the minimizer.

The authors consider convergence of the JKO-like scheme
$$
\rho_{n+1} \in \mathrm{prox}_{F,\xi}(\rho_n) :=  \arg \min\_{\rho \in \mathcal{P}_2(\Theta)}   F(\rho) + \frac{1}{2\xi}W_2^2 (\rho, \rho_n)\,.
$$
Under some natural and some strong assumptions (more on this later) the authors prove linear convergence of this scheme (Theorem 3.5) and claim to verify assumptions in some specific cases.
These are the MFLD - Corollary 3.7 and then Corollary 3.9 for $F$ which is neither flat nor geodesically convex.

The natural assumption is that $F$ satisfies the Wasserstein Polyak-Lojasiewicz (PL) inequality condition.
Existing results [2] and those mentioned in the paper under review assume either geodesic convexity of $F$ (or the closely related evolutional variational inequality (EVI)) or flat convexity [3] and then proceed via the log-Sobolev inequality (LSI) and the entropy sandwich, see [3].

The first strong assumption is Assumption 1, part (1) namely that their scheme admits a minimizer which is moreover absolutely continuous with respect to the Lebesgue measure on $\Theta$.
For compact theta this becomes especially problematic: how do we know that the minimizer does not accumulate mass on the boundary?

The second strong assumption is Assumption 2 which rougly speaking says that we can relate the Wasserstein derivative of $F$ at any $\rho_\xi \in  \text{prox}\_{F,\xi}(\rho)$ for any $\rho$ in the domain of $F$ and the optimal transport from $\rho_\xi$ to $\rho$.

Finally, the authors consider the case of inexact resolution of the proximal scheme in terms of approximate transport maps.
They show, in Theorem 3.12, that with inexact resolution of the scheme the convergence remains linear but only up to an error term arising from the approximation.

### References

[1] L. Ambrosio, N. Gigli, and G. Savare. Gradient Flows: In Metric Spaces and in the Space of Probability Measures. Lectures in Mathematics. ETH Zurich. Birkhauser Basel, 2008.

[2] A. Salim, A. Korba, and G. Luise. The Wasserstein proximal gradient algorithm. In Advances in Neural Information Processing Systems, volume 33, pages 12356–12366., 2020.

[3] Lascu, R. A., Majka, M. B., Šiška, D., & Szpruch, Ł. (2024). Linear convergence of proximal descent schemes on the Wasserstein space. arXiv preprint arXiv:2411.15067.

**Additional Comments:**

Ideally, a revision which resolves the major comments will undergo another round of reviews. This reviewer isn't sure exactly what the process is with TMLR but cannot recommend acceptance of the current version.

**Audience:**

Yes

**Audience Explanation:**

Researchers working on optimal transport, mean-field optimization, and the theory of sampling/generative modeling will find the convergence bounds and the analysis of the inexact proximal algorithm interesting, provided the context of existing literature is properly established.

**Claims And Evidence:**

No

**Claims Explanation:**

Not yet.

While it is clear that the authors have been careful when writing their proofs a few issues remain. Resolving the major objections in the requested changes, mainly A2, will easily make this into a "yes".

**Requested Changes:**

### Major

A1: There is lack of clarity about whether $\Theta$ is meant to be a compact subset of $\mathbb R^d$ or in fact $\mathbb R^d$ itself or something else.

Already after (1) we encounter the unusual statement "on a non-compact domain $\Theta \in \mathbb R^d$".
Such $\Theta$ of course exists e.g. $[0,\infty) \subset \mathbb R^d$ but they represent analytically the hardest setting: we have a boundary to worry about and we don't have compactness.

Then discussion around (5) and (6) implicitly imply $\Theta = \mathbb R^d$ (since the diffusions immediately leave any compact subset of $\mathbb R^d$).

A2: Incomplete Proof of Lemma B.2 (Existence of Minimizer). The proof provided for Lemma B.2 (which justifies Assumption 1) is incomplete.
The authors state that they are providing a proof for the sake of completeness, but the text only establishes that bounded Wasserstein balls are weakly precompact (tight).
It entirely omits the actual optimization argument where the functional $F$ is involved.
Also, how is this supposed to work without assuming that $F$ is bounded from below?

A3: The submission completely misses highly relevant and recent paper [3].
In particular [3] tackles a similar objective: establishing linear convergence for JKO-based schemes without assuming geodesic convexity, specifically applying it to the same two-layer mean-field neural network problem.
The authors should explicitly detail their theoretical contributions over and above it (e.g., their analysis of the inexact proximal algorithm).
It also seems that when veryfing that the setting of the paper under review is applicable, mostly tools, which already appeared in [3], have been used (entropy sandwich, LSI).

### Minor

B1: The statement "the (geodesically convex) negative entropy functional Ent($\rho$) = $\int \rho \log \rho$  is not Wasserstein differentiable (see Appendix A for definitions) for any $\rho \in \mathcal P^a_2(\mathbb R^d)$ such that Ent($\rho$) $< \infty$" is of course accurate but may be misleading to those new in the field: if we have additional regularity of the density then the entropy has unique Wasserstein subdifferential c.f. Theorem 10.4.9 in [1]. I.e. entropy is not "everywhere Wasserstein differentiable" but it is very far from being "nowhere Wasserstein Differentiable".

B2: In the same way that the authors distinguish between $<$ and $\leq$ they should distinguish between $\subset$ and $\subseteq$.
For example one wouldn't write $(-\infty,\infty) \subset \mathbb R$ but rather $(-\infty,\infty) \subseteq \mathbb R$.

B3: Absolute continuity between measures requires two measures. So when the authors write "be the subset of absolutely continuous measures" they should specify absolutely continuous relative to what (presumably Lebesgue but why let the reader guess).

B4: The definition of first variation (Definition A.2) is not valid because the authors use a linear perturbation but the domain of $F$ is only convex.
Also $\mu$ appears but is undefined, this reviewer assumed this is meant to be $\rho$.